# RainProof: An umbrella to shield text generators from Out-Of-Distribution data

## Abstract

As more and more conversational and translation systems are deployed in production, it is essential to implement and to develop effective control mechanisms guaranteeing their proper functioning and security. An essential component to ensure safe system behavior is out-of-distribution (OOD) detection, which aims at detecting whether an input sample is statistically far from the training distribution. Although OOD detection is a widely covered topic in classification tasks, it has received much less attention in text generation. This paper addresses the problem of OOD detection for machine translation and dialog generation from an operational perspective. Our contributions include: (i) RAINPROOF a Relative informAItioN Projection ODD detection framework; and (ii) a more operational evaluation setting for OOD detection. Surprisingly, we find that OOD detection is not necessarily aligned with task-specific measures. The OOD detector may filter out samples that are well processed by the model and keep samples that are not, leading to weaker performance. Our results show that RAINPROOF breaks this curse and achieve good results in OOD detection while increasing performance.

## 1 Introduction

Significant progress have been made in Natural Language Generation (NLG) in recent years with the development of powerful generic (*e.g.,* GPT (Radford et al., 2018; 2019; Brown et al., 2020)) and task-specific (*e.g.,* Grover (Zellers et al., 2019), Pegasus (Zhang et al., 2020) and DialogGPT (Zhang et al., 2019)) text generators. Text generators power machine translation systems or chat bots that are by definition exposed to the public and whose reliability is therefore a prerequisite for adoption. Text generators are trained in the context of a so-called closed world (Antonucci et al., 2021; Fei & Liu, 2016), where training and test data are assumed to be drawn i.i.d. from a single distribution, known as the in-distribution. However, when deployed, these models operate in an open world (Parmar et al., 2021; Zhou, 2022) where the i.i.d. assumption is often violated. This change in data distribution is detrimental and induces a drop in performance as illustrated in Tab. 3 and Tab. 4. Thus, to ensure the trustworthiness and adoption, it is necessary to develop tools to protect them from harmful distribution shifts. For example, a trained translation model is not expected to be reliable when presented with another language (*e.g.* a Spanish model exposed to Catalan, or a Dutch model exposed to Afrikaans) or unexpected technical language (*e.g.,* a colloquial translation model exposed to rare technical terms from the medical field).

Most of the existing research, which aims to protect models from Out-Of-Distribution (OOD) data, focuses on classification. Despite their importance, (conditional) text generation has received much less attention even though it is among the most exposed applications. Existing solutions fall into two categories. The first one called training-aware methods (Zhu et al., 2022; Vernekar et al., 2019a;b) modifies the classifier training by exposing the neural network to OOD samples during training. The second one called plug-in methods aims at distinguishing regular samples in the in distribution (IN) from OOD samples based on the behavior of the model on a new input. Plug-in methods include Maximum Softmax Prediction (MSP) (Hendrycks & Gimpel, 2016) or Energy (Lee et al., 2018a) or feature-based anomaly detectors that compute a per-class anomaly score (Ming et al., 2022; Ryu et al., 2017; Huang et al., 2020; Ren et al., 2021a). Although plug-in methods seem attractive, their adaptation to text generation may not be straightforward. The sheer number of words present in the vocabulary prevents it to be used directly within the classification framework.

In this work, we aim at developing new tools to build more reliable text generators, which can be used in practical systems. First, we work in the *unsupervised* detection setting where we do not assume that we have access to OOD samples as they are often not available. Second, we work in the *black-box scenario*, which is the most common in the Software as a Service framework Rudin & Radin (2019). In the black-box setting detection methods only have access to the output of the DNN architecture. Third, we want an *easy-to-use and effective method to ensure adoptability*. Last, we argue that OOD detection impacts on *tasks specific performance* of the whole system should be taken into account when choosing OOD detectors in an operational setting.

**Our contributions.** Our main contributions can be summarized as follows:

1. *A more operational benchmark for text generation OOD detection.* We present LOFTER the **L**anguage **O**ut o**F** dis**T**ribution p**E**rformance benchma**R**k. Existing works on OOD detection for language modeling (Arora et al., 2021) focus on (i) english language only, (ii) the GLUE benchmark and (iii) measure performance solely in terms of OOD detection. LOFTER is, in our view, a more operational setting with a strong focus on neural machine translation (NMT) and dialog generation. First, it introduces more realistic data shifts that go beyond English Fan et al. (2021): language shifts induced by closely related language pairs (*e.g.*, Spanish and Catalan or Dutch and Afrikaans[1]) and domain change (*e.g.*, medical vs news data or different types of dialogs). In addition, LOFTER comes with an updated evaluation setting: detectors' performance are jointly evaluated w.r.t the overall system's performance on the end task.

2. *Novel information theoretic-based detectors.* We present RAINPROOF: a **R**elative inform**AI**tio**N** **P**rojection **O**ut **OF** distribution detector. RAINPROOF is fully **unsupervised**. It is flexible and can be applied both when no reference samples (IN) are available (corresponding to scenario $s_0$) and when they are (corresponding to scenario $s_1$). RAINPROOF tackles $s_0$ by computing the models' predictions negentropy (Brillouin, 1953). For $s_1$, it relies its natural extension: the Information Projection (Kullback, 1954; Csiszár, 1967), an information-theoretic tool that remains overlooked by the machine learning community.

3. *New insights on the operational value of OOD detectors* Our extensive experiments on LOFTER show that *OOD detectors may filter out samples that are well processed by the model and keep samples that are not, leading to weaker performance.* Our results show that RAINPROOF breaks this curse and achieve good results in OOD detection while increasing performance.

4. *Code and reproducibility.* After acceptance, we will publish the open-source code on `github.com` and the data to facilitate future research, ensure reproducibility and reduce computational costs.

## 2 PROBLEM STATEMENT & RELATED WORKS

### 2.1 NOTATIONS & CONDITIONAL TEXT GENERATION

Let us denote $\Omega$ a vocabulary of size $|\Omega|$ and $\Omega^*$ its Kleene closure (Fletcher et al., 1990)[2]. We denote $\mathcal{P}(\Omega) = \left\{ \mathbf{p} \in [0,1]^{|\Omega|} : \sum_{i=1}^{|\Omega|} \mathbf{p}_i = 1 \right\}$ the set of probability distributions defined over $\Omega$. Let $\mathcal{D}_{train}$ be the training set, composed of $N \geqslant 1$ i.i.d. samples $\{(\mathbf{x}^i, \mathbf{y}^i)\}_{i=1}^N \in (\mathcal{X} \times \mathcal{Y})^N$ with probability law $\mathbf{p}_{XY}$. We denote $\mathbf{p}_X$ and $\mathbf{p}_Y$ the associated marginal laws of $\mathbf{p}_{XY}$. Each $\mathbf{x}^i$ is a sequence of tokens and we denote $x_j^i \in \Omega$ the $j$th token of the $i$th sequence. $\mathbf{x}_{\leqslant t}^i = \{x_1^i, \cdots, x_t^i\} \in \Omega^*$ denotes the prefix of length $t$. The same notations hold for $\mathbf{y}$.

**Conditional textual generation.** In conditional textual generation, the goal is to model a probability distribution $\mathbf{p}_\star(\mathbf{x}, \mathbf{y})$ over variable-length text sequences $(\mathbf{x}, \mathbf{y})$ by finding $\mathbf{p}_\theta \approx \mathbf{p}_\star(\mathbf{x}, \mathbf{y})$ for any $(\mathbf{x}, \mathbf{y})$. In this work, we assume to have access to a pretrained conditional language model $f_\theta : \mathcal{X} \times \mathcal{Y} \to \mathbf{R}^{|\Omega|}$ where the output is the (unnormalized) logits scores. $f_\theta$ parameterized $\mathbf{p}_\theta$, *i.e.,* for any $(\mathbf{x}, \mathbf{y})$, $\mathbf{p}_\theta(\mathbf{x}, \mathbf{y}) = \mathrm{softmax}(f_\theta(\mathbf{x}, \mathbf{y})/T)$ where $T \in \mathbf{R}$ denotes the temperature. Given an input sequence $\mathbf{x}$, the pretrained language $f_\theta$ can recursively generate an output sequence $\hat{\mathbf{y}}$ by

---

[1] Afrikaans is a daughter language of Dutch (Jansen et al., 2007). The Dutch sentence: Appelen zijn gewoonlijk groen, geel of rood can be translated in "Appels is gewoonlik groen, geel of rooi."

[2] The Kleene closure corresponds to sequences of arbitrary size written with words in $\Omega$. Formally: $\Omega^* = \bigcup_{i=0}^{\infty} \Omega^i$.

sampling $y_{t+1} \sim \mathbf{p}_\theta^T(\cdot|\mathbf{x}, \hat{\mathbf{y}}_{\leqslant t})$, for $t \in [1, |\mathbf{y}|]$. Note that $\hat{y}_0$ is the start of sentence ($< \mathrm{SOS} >$ token). We denote by $\mathcal{S}(\mathbf{x})$, the set of normalized logits scores generated by the model when the initial input is $\mathbf{x}$ *i.e.*, $\mathcal{S}(\mathbf{x}) = \{\mathrm{softmax}(f_\theta(\mathbf{x}, \hat{\mathbf{y}}_{\leqslant t}))\}_{t=1}^{|\hat{\mathbf{y}}|}$. Note that elements of $\mathcal{S}(\mathbf{x})$ are discrete probability distributions on $\Omega$.

## 2.2 PROBLEM STATEMENT

In OOD detection the goal is to find an anomaly score $a : \mathcal{X} \to \mathbf{R}_+$ that quantifies how much a sample is far from the IN distribution. $\mathbf{x}$ is classified as IN or OUT according to the score $a(\mathbf{x})$. Following previous work (Hendrycks & Gimpel, 2016), one fixes a threshold $\gamma$ and classifies the test sample IN if $a(\mathbf{x}) \leqslant \gamma$ or OOT if $a(\mathbf{x}) > \gamma$. Formally, let us denote $g(\cdot, \gamma)$ the decision function, we take: $g(\mathbf{x}, \gamma) = \begin{cases} 1 & \text{if } a(\mathbf{x}) > \gamma \\ 0 & \text{if } a(\mathbf{x}) \leqslant \gamma \end{cases}$

*Remark* 1. *In our setting, OOD examples are not available*. In our experiments, we take $\gamma$ such that at least $80\%$ of the train set is classified as IN data. This assumption is reasonable since, in practice, even a well tailored dataset might contains significant shares of outliers (Mishra et al., 2020).

## 2.3 REVIEW OF OOD DETECTORS

**OOD detection for classification.** Most works on OOD detection have focused on detectors for classifiers and relies either on internal representations (*features-based detectors*) or on the final soft probabilities produced by the classifier (*softmax based detectors*).

**Features-based detectors.** They leverage latent representations to derive anomaly scores (Kirichenko et al., 2020; Zisselman & Tamar, 2020). The most well-known is the Mahanalobis distance (Lee et al., 2018b; Ren et al., 2021b) but there are other methods employing Grams matrices (Sastry & Oore, 2020), Fisher Rao distance (Gomes et al., 2022) or other statistical tests (Haroush et al., 2021). Other methods rely on the gradient space (Huang et al., 2021) or the moment of the features (Quintanilha et al., 2019; Sun et al., 2021). These methods require access to the latent representations of the models, which does not fit the black-box scenario. Moreover, they often rely on a per-class decision, which is fine for classifiers but the sheer number of words in $\Omega$ makes it impossible to use for text generation.

**Softmax-based detectors.** These detectors rely on the soft probabilities produced by the model. The maximum softmax probability (Hendrycks & Gimpel, 2017; Hein et al., 2019; Liang et al., 2018; Hsu et al., 2020) uses the probability of the mode while others take into account the entire distribution, such as the Energy-based OOD detection scores (Liu et al., 2020). Due to the large vocabulary size, it is unclear how these methods generalize to sequence generation tasks.

**OOD detection for text generation.** Little work has been done on OOD detection for text generation. Therefore, we will follow Arora et al. (2021) and will rely on their baselines but also generalize common OOD scores such as MSP or Energy to the context of text generation.

**Generalization to sequence generation.** We generalize common OOD detectors for classification tasks by computing the average OOD score along the sequence at each step of the text generation. We refer the reader to Sec. A.6 for more details.

*Remark* 2. Note that *features-based* detectors assume a white-box framework where the internal representations of an input are accessible. By contrast to *softmax-based* detectors which only rely on the final output. Following Arora et al. (2021), we work in a black-box framework (Chen et al., 2020). We also compare our results to the Mahalanobis distance (Lee et al., 2018b), as it is known to be a strong baseline.

## 3 RAINPROOF AN INFORMATION THEORETIC OOD DETECTORS

### 3.1 INFORMATION THEORETICAL BACKGROUND

An information measure $\mathcal{I} : \mathcal{P}(\Omega) \times \mathcal{P}(\Omega) \to \mathbf{R}$ quantifies the similarity between any pair of discrete distributions $\mathbf{p}, \mathbf{q} \in \mathcal{P}(\Omega)$. Since $\Omega$ is a finite set, we will adopt the following notations $\mathbf{p} = [\mathbf{p}_1, \cdots, \mathbf{p}_{|\Omega|}]$ and $\mathbf{q} = [\mathbf{q}_1, \cdots, \mathbf{q}_{|\Omega|}]$. The development of new information measures for

specific applications has received much attention over the years (Fujisawa & Eguchi, 2008; Cichocki et al., 2011) (we refer the reader to Basseville (2013) for a complete review). While there exist information distances, it is, in general, difficult to build metrics that satisfy all the properties of a distance, thus we often rely on divergences which drop the symmetry property and the triangular inequality. In what follows, we motivate the information measures we will use in this work.

First, we rely on the *Rényi divergences* (Csiszár, 1967). Rényi divergences belong to the $f$-divergences family and are parametrized by a parameter $\alpha \in \mathbf{R}_+ - \{1\}$. They are flexible and include well-known divergences such as the Kullback-Leiber divergence (KL) Kullback (1959) (when $\alpha \to 1$) or the Hellinger distance (Hellinger, 1909) (when $\alpha = 0.5$). The Rényi divergence between $\mathbf{p}$ and $\mathbf{q}$ is defined as follows:

$$D_\alpha(\mathbf{p}\|\mathbf{q}) = \frac{1}{\alpha - 1} \log \left( \sum_{i=1}^{|\Omega|} \frac{\mathbf{p}_i^\alpha}{\mathbf{q}_i^{\alpha-1}} \right). \tag{1}$$

The Renyi divergence is widely used in machine learning (Peters et al., 2019) because $\alpha$ allows weighting the relative influence of the distributions' tail.

Second, we investigate the *Fisher-Rao distance* (FR). FR is a distance on the Riemannian space formed by the parametric distributions, using Fisher information matrix as its metric (Amari, 2012). It computes the geodesic distance between two discrete distributions (Rao, 1992; Pinele et al., 2020) and is defined as follows:

$$\mathrm{FR}(\mathbf{p}\|\mathbf{q}) = \frac{2}{\pi} \arccos \sum_{i=1}^{|\Omega|} \sqrt{\mathbf{p}_i \times \mathbf{q}_i}. \tag{2}$$

It has recently found many applications (Picot et al., 2022; Colombo et al., 2022b;a) and is known to be more accurate than popular divergence measures (Costa et al., 2015).

## 3.2 RAINPROOF FOR THE NO-REFERENCE SCENARIO ($\mathbb{s}_0$)

At inference time, the no-reference scenario ($\mathbb{s}_0$) does not assume the existence of a reference set of IN samples to decide whether a new input sample is OOD. Softmax-based detectors such as MSP (Hendrycks & Gimpel, 2016), Energy (Liu et al., 2020) or the sequence likelihood[3] (Arora et al., 2021) are examples of OOD scores operating under $\mathbb{s}_0$.

Under these assumptions, our OOD detector RAINPROOF is composed of three steps. For a given input $\mathbf{x}$ with generated sentence $\hat{\mathbf{y}}$:

1. We first use $f_\theta$ to extract the step-by-step sequence of soft distributions $\mathcal{S}(\mathbf{x})$.

2. We then compute an anomaly score ($a_\mathcal{I}(\mathbf{x})$) by averaging a step-by-step score provided by $\mathcal{I}$. This step-by-step score is obtained by measuring the similarity between a reference distribution $\mathbf{u} \in \mathcal{P}(\Omega)$ and one element of $\mathcal{S}(\mathbf{x})$. Formally:

$$a_\mathcal{I}(\mathbf{x}) = \frac{1}{|\mathcal{S}(\mathbf{x})|} \sum_{\mathbf{p} \in \mathcal{S}(\mathbf{x})} \mathcal{I}(\mathbf{p}\|\mathbf{u}), \tag{3}$$

where $|\mathcal{S}(\mathbf{x})| = |\hat{\mathbf{y}}|$.

3. The last step consists in thresholding the previous anomaly score $a_\mathcal{I}(\mathbf{x})$. If $a_\mathcal{I}(\mathbf{x})$ is over a given threshold $\gamma$, we classify $\mathbf{x}$ as an OOD example.

**Interpretation of Eq. 3.** $a_\mathcal{I}(\mathbf{x})$ measures the average dissimilarity of the probability distribution of the next token to normality (as defined by $\mathbf{u}$). $a_\mathcal{I}(\mathbf{x})$ also corresponds to the token average uncertainty of the model $f_\theta$ to generate $\hat{\mathbf{y}}$ when the input is $\mathbf{x}$. The intuition behind Eq. 3 is that the distributions produced by $f_\theta$, when exposed to an OOD sample, should be far from normality and thus should have a high score.

**Choice of $\mathbf{u}$ and $\mathcal{I}$.** The uncertainty definition of Eq. 3 depends on the choice of both the reference distribution $\mathbf{u}$ and the information measure $\mathcal{I}$. A natural choice for $\mathbf{u}$ is the uniform distribution,

---

[3]The likelihood of the sequence is the same as the perplexity. In our work we report the log-likelihood for numerical stability reasons: *i.e.,* $a_L(\mathbf{x}) = -\sum_{t=0}^{|\hat{\mathbf{y}}|-1} \log \mathbf{p}_\theta(\hat{\mathbf{y}}_{t+1}|\mathbf{x}, \hat{\mathbf{y}}_{\leqslant t})$

*i.e.,* $\mathbf{u} = [\frac{1}{|\Omega|}, \cdots, \frac{1}{|\Omega|}]$ which we will use in this work. It is worth pointing out that $\mathcal{I}(\cdot \| \mathbf{u})$ yields the negentropy of a distribution. Other possible choices for $\mathbf{u}$ include one hot or tf-idf distribution (Colombo et al., 2022b). For $\mathcal{I}$, we rely on the Rényi divergence to obtain $a_{\mathcal{D}_\alpha}$ and the Fisher-Rao distance to obtain $a_{\text{FR}}$.

### 3.3 RAINPROOF FOR THE REFERENCE SCENARIO ($\mathtt{s}_1$)

In the ~~with~~ reference scenario ($\mathtt{s}_1$), we assume that one has access to a reference set of **IN samples** $\mathcal{R} = \{\mathbf{x}^i : (\mathbf{x}^i, \mathbf{y}^i) \in \mathcal{D}_{train}\}_{i=1}^{|\mathcal{R}|}$ where $|\mathcal{R}|$ is the size of the reference set. For example, the Mahalanobis distance works under this assumption. One of the weakness of Eq. 3 is that it imposes ~~is to imposes~~ an ad-hoc choice when using $\mathbf{u}$ (the uniform distribution). In $\mathtt{s}_1$, we can leverage $\mathcal{R}$, to obtain a *data-driven notion normality*.

Under $\mathtt{s}_1$, our OOD detector RAINPROOF follows these four steps:

1. (*Offline*) For each $\mathbf{x}^i \in \mathcal{R}$, we generate $\hat{\mathbf{y}}^i$ and the associated sequence of probability distributions ($\mathcal{S}(\mathbf{x}^i)$). Overall we thus generate $\sum_{\mathbf{x} \in \mathcal{R}} |\hat{\mathbf{y}}^\mathbf{i}|$ probability distributions which could explode for long sequences[4]. To overcome this limitation, we rely on the bag of distributions of each sequence (Colombo et al., 2022b). We form the set of these bags of distributions

$$\bar{\mathcal{S}}^* = \bigcup_{\mathbf{x}^i \in \mathcal{R}} \left\{ \frac{1}{|\mathcal{S}(\mathbf{x}^i)|} \sum_{\mathbf{p} \in \mathcal{S}(\mathbf{x}^i)} \mathbf{p} \right\}. \tag{4}$$

2. (*Online*) For a given input $\mathbf{x}$ with generated sentence $\hat{\mathbf{y}}$, we compute its bag of distributions representation

$$\bar{\mathbf{p}}(\mathbf{x}) = \frac{1}{|\mathcal{S}(\mathbf{x})|} \sum_{\mathbf{p} \in \mathcal{S}(\mathbf{x})} \mathbf{p}. \tag{5}$$

3. (*Online*) For $\mathbf{x}$, we then compute an anomaly score $a_{\mathcal{I}}^\star(\mathbf{x})$ by projecting $\bar{\mathbf{p}}(\mathbf{x})$ on the set $\bar{\mathcal{S}}^*$. Formally, $a_{\mathcal{I}}^\star(\mathbf{x})$ is defined as:

$$a_{\mathcal{I}}^\star(\mathbf{x}) = \min_{\mathbf{p} \in \bar{\mathcal{S}}^\star} \mathcal{I}(\mathbf{p} \| \bar{\mathbf{p}}(\mathbf{x})). \tag{6}$$

We denote $\mathbf{p}^\star(\mathbf{x}) = \arg\min_{\mathbf{p} \in \bar{\mathcal{S}}^*} \mathcal{I}(\mathbf{p} \| \bar{\mathbf{p}}(\mathbf{x}))$.

4. The last step consists of thresholding the previous anomaly score $a_{\mathcal{I}}(\mathbf{x})$. If $a_{\mathcal{I}}(\mathbf{x})$ is over a given threshold $\gamma$, we classify $\mathbf{x}$ as an OOD example.

**Interpretation of Eq. 6.** $a_{\mathcal{I}}(\mathbf{x})$ relies on a Generalized Information Projection (Kullback, 1954; Csiszár, 1975; 1984)[5] which measures the similarity between $\bar{\mathbf{p}}(\mathbf{x})$ and the set $\bar{\mathcal{S}}^*$. Note that the closest element of $\bar{\mathcal{S}}^*$ in the sens of $\mathcal{I}$ can give insights on the decision of the detector. It allows to interpret the decision of the detector as we will see in Tab. 5.

**Choice of $\mathcal{I}$.** Similarly to Sec. 3.2, we will rely on the Rényi divergence to define $a_{\mathcal{R}_\alpha}^\star(\mathbf{x})$ and the Fisher-Rao distance $a_{\text{FR}}^\star(\mathbf{x})$.

## 4 RESULTS ON LOFTER

### 4.1 LOFTER: LANGUAGE OUT OF DISTRIBUTION PERFORMANCE BENCHMARK

**LOFTER for NMT.** We consider two main types of changes: language changes and domain changes, which both can occur in real-world situations. For each shift, we rely on pretrained generators from the HuggingFace Hub. Further experimental details are relegated to Ap. A. *Language shifts* can

---

[4]It is also worth pointing that doing a projection at each timestep would require a per-step reference set in addition to the computational time required to actually compute the projections, therefore we decided to aggregate the probability distributions over the sequence.

[5]The minimization problem of Eq. 6 finds numerous connections in the theory of large deviation (Sanov, 1958) or in statistical physics (Jaynes, 1957).

Table 1: Summary of the performance and computational cost of every detector.

(a) Summary of the performance of our detectors (Ours) compared to commonly used strong baselines (Bas.). We report in bold the best detector for each scenario and we underline the best overall.

|  |  |  | Language shifts | | | Domain shifts | | | Dialog shifts | | |
|---|---|---|---|---|---|---|---|---|---|---|---|
|  |  |  | AUROC | FPR | F1 | AUROC | FPR | F1 | AUROC | FPR | F1 |
| $S_0$ | Ours | $a_{D_\alpha}$ | **0.95** | **0.25** | **0.84** | **0.85** | 0.62 | **0.75** | 0.79 | 0.64 | **0.66** |
|  |  | $a_{\mathrm{FR}}$ | 0.93 | 0.28 | 0.83 | 0.74 | 0.87 | 0.60 | 0.72 | 0.70 | 0.64 |
|  | Bas. | $a_E$ | 0.89 | 0.44 | 0.77 | 0.76 | 0.78 | 0.71 | 0.65 | 0.76 | 0.57 |
|  |  | $a_{\mathrm{MSP}}$ | 0.87 | 0.44 | 0.75 | 0.78 | 0.77 | 0.71 | 0.66 | 0.72 | 0.21 |
|  |  | $a_L$ | 0.78 | 0.79 | 0.50 | 0.72 | 0.89 | 0.65 | 0.65 | 0.95 | 0.62 |
| $S_1$ | Ours | $a_{D_\alpha^*}$ | 0.88 | 0.34 | 0.78 | **0.86** | **0.50** | **0.70** | **0.86** | **0.52** | **0.59** |
|  |  | $a_{\mathrm{FR}^*}$ | 0.88 | 0.35 | 0.77 | 0.81 | 0.69 | 0.69 | 0.76 | 0.75 | 0.38 |
|  | Bas. | $a_M$ | **0.92** | **0.26** | **0.73** | 0.78 | 0.59 | 0.40 | 0.84 | 0.55 | 0.56 |
|  |  | $a_C$ | 0.71 | 0.80 | 0.62 | 0.68 | 0.76 | 0.67 | 0.72 | 0.61 | 0.48 |

(b) Computation time (in seconds) for the different detectors. Off. (Onl.) stands for offline (resp. online) time.

| Score | Off. | Onl. |
|---|---|---|
| $a_{D_\alpha}$ | ✗ | $2.10^{-3}$ s |
| $a_{\mathrm{MSP}}$ | ✗ | $1.10^{-4}$ s |
| $a_M$ | 40s | $3.10^{-3}$ s |
| $a_{D_\alpha^*}$ | ✗ | $9.10^{-2}$ s |

appear when a translation system is exposed to a language that is extremely similar to the language the system has been trained on (*e.g.*, Afrikaans for a system trained on Dutch) and, therefore, can lead to significant translation errors (see Tab. 7)). For language shifts, we focus on closely related language pairs coming from the Tatoeba dataset (Tiedemann, 2012b) (see Tab. 6). We study the shifts induced by Catalan-Spanish, Portugese-Spanish and Afrikaans-Dutch. *Domain shifts*, which occur when the model is exposed to a specific topic that was not seen during training, can also affect the quality of the translation (see Tab. 4). To simulate domain shifts, we use the language Tatoeba MT dataset (Tiedemann, 2020) and the news commentary dataset (Tiedemann, 2012b) as base datasets and the shifts are induced by the EuroParl dataset (Tiedemann, 2012a) and EMEA (Tiedemann, 2012b) dataset.

**LOFTER for dialogs.** For conversational agents, an interesting scenario is when a goal-oriented agent designed to handle a specific type of conversations (*e.g.*, customer conversations, daily dialogue) is exposed to an unexpected conversation. In this case, it is crucial to interrupt the agent so it does not damage the user's trust with misplaced responses (Perez et al., 2022). We rely on the Multi WOZ dataset (Zang et al., 2020), a human to human dataset collected in the Wizard-of-Oz set-up (Kelley, 1984), for IN distribution data. This choice is mostly motivated by the availability of pretrained models on Multi WOZ. For dialog shifts, we use spoken datasets coming from various sources which are part of the SILICONE benchmark (Chapuis et al., 2020). Specifically, we use a goal-oriented dataset (*i.e.*, Switchboard Dialog Act Corpus (SwDA) (Stolcke et al., 2000)), a multi-party meetings dataset (*i.e.*, MRDA (Shriberg et al., 2004) and Multimodal EmotionLines Dataset MELD (Poria et al., 2018)), daily communication dialogs ( *i.e.*, DailyDialog DyDA Li et al. (2017)), and scripted scenarii (*i.e.*, IEMOCAP Tripathi et al. (2018)). We refer the curious reader to Sec. A.4 for more details on each dataset.

**Metrics.** OOD detection is usually framed as an unbalanced binary classification problem where the class of interest is OUT. We can assess the performance of our OOD detectors focusing on the **False alarm rate** (FPR) and on the **True detection rate** (TPR). To evaluate the performance on the OOD task we report the AUROC and the FPR.
*Area Under the Receiver Operating Characteristic curve (AUROC) (Bradley, 1997)*. The AUROC can be interpreted as the probability that an IN-distribution example has an higher anomaly score than an OOD sample. For this metric, higher is better.
*False Positive Rate at $r\%$ True Positive Rate (FPR)*. In many practical application, we have to detect at least $r\%$ of the the OOD samples. This corresponds to pre-defined safety level. FPR quantifies the share of IN samples we wrongly detect under this constraint. It leads to select a threshold $\gamma_r$ such that the corresponding TPR equals $r$. In our work $r$ is set to $95\%$. Additional details on these metrics can be found in Sec. A.1. **F1, precision and recall**. In addition we report the F1 scores of the detectors with a threshold designed such that $80\%$ of the IN dataset is actually classified as IN.

## 4.2 EXPERIMENTS IN MACHINE TRANSLATION AND RESULTS

**Results on language shifts.** We assess, for each language pair, the OOD detection performance of RAINPROOF and report the average AUROC and FPR in Tab. 1a. We provide the detailed results in Tab. 8. We find that our no-reference methods ($a_{D_\alpha}$ and $a_{\mathrm{FR}}$) achieve better performance that

common no-reference baselines but also outperform the reference-based baseline. In particular, $a_{D_\alpha}$, by achieving an AUROC of $0.95$ and FPR of $0.25$, outperforms all considered methods. Moreover, while no-reference baselines only capture up to $62\%$ of the OOD samples on average, ours detect up to $83.5\%$, achieving even better results than the with-reference baseline ($75.3\%$).

**Results on domain shifts.** We evaluate the OOD detection performance of RAINPROOF on domain shifts in Spanish and German with technical medical data and parliamentary data. We report the average OOD detection performance in Tab. 1a. In $\mathbb{s}_0$, we observe that $a_{D_\alpha}$ and $a_{\text{FR}}$ outperform the strongest baselines (*i.e.*, Energy, MSP and sequence likelihood) by several AUROC points. Interestingly enough even our no-reference detectors outperform the reference-based baseline (*i.e.*, $a_M$). However, we find that relying on a reference set

Table 2: Correlation between OOD scores and translation metrics BLEU and BERT-S on domain shifts datasets.

| Scenario | | Score | Bertscore f1 | | | Bleu score | | |
|---|---|---|---|---|---|---|---|---|
| | | | ALL | IN | OUT | ALL | IN | OUT |
| $\mathbb{s}_0$ | Ours | $a_{D_\alpha}$ | -0.29 | -0.26 | -0.18 | -0.17 | -0.22 | -0.09 |
| | | $a_{\text{FR}}$ | -0.36 | -0.30 | -0.27 | -0.24 | -0.26 | -0.19 |
| | Baselines | $a_E$ | -0.19 | -0.24 | -0.33 | -0.26 | -0.19 | -0.39 |
| | | $a_L$ | -0.47 | -0.51 | -0.48 | -0.49 | -0.45 | -0.49 |
| | | $a_{\text{MSP}}$ | -0.15 | -0.19 | -0.29 | -0.24 | -0.16 | -0.37 |
| $\mathbb{s}_1$ | Ours | $a_{D_\alpha^*}$ | -0.12 | 0.00 | -0.09 | -0.19 | 0.00 | -0.09 |
| | | $a_{\text{FR}^*}$ | -0.11 | 0.00 | -0.12 | -0.17 | 0.00 | -0.10 |
| | Baselines | $a_C$ | -0.02 | -0.04 | 0.01 | -0.13 | -0.05 | -0.08 |
| | | $a_M$ | 0.02 | 0.00 | 0.06 | -0.06 | 0.00 | 0.07 |

is a must-have in terms of FPR. While $a_{D_\alpha}$ achieves similar AUROC performance to its information projection counterpart $a_{D_\alpha^*}$, the latter achieve much better FPR.

### 4.3 EXPERIMENTS IN DIALOG GENERATION AND RESULTS

**Results on Dialog shifts.** The dialog shifts benchmark is more difficult than NMT benchmark as all detectors achieve lower performances. It is the only case where our no-reference detectors do not outperform the Mahalanobis baseline and achieve only $0.79$ in AUROC. The best baseline is the Mahalanobis distance and achieves better performance on dialog task than on NMT domain shifts reaching an AUROC of $0.84$. However, our reference based detector based on the Rényi information projection secures better AUROC ($0.86$) and better FPR ($0.52$). Even though RAINPROOF outperforms all the baselines, shifts in dialog are hard to detect and will require further investigations. Non-aggregated results for dialog are provided in Ap. C. They show that RAINPROOF consistently outperforms baselines on all datasets.

**Importance of distribution tails.** Our results show that, when it comes to domain shift (domain shifts in translation or dialog shifts), reference-based detectors are required to obtain good results. They also show that, the more these detectors take into account the tail of the distributions, the better they are, as displayed in Sec. B.1. We find that low values of $\alpha$ (near 0) yields better results with the Rényi Information projection $a_{D_\alpha^*}$. It suggests that the tail of the distributions used during text generation carries context information and insights on the processed texts. Such results are consistent with findings of recent works in the context of automatic evaluation of text generation (Colombo et al., 2022b).

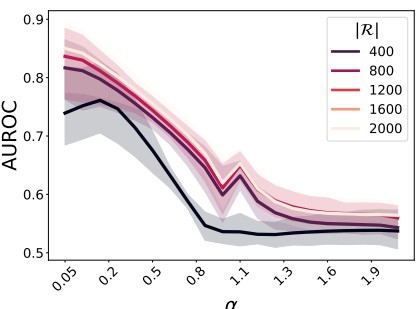

Figure 1: Impact of $\alpha$ on the performance of the Rényi information projection for dialog shifts detection. A smaller $\alpha$ increases the weight of the tail of the distribution. An $\alpha$ of 0 would consist in counting the number of the common non zero elements.

**Comparison to the Mahalanobis distance.** Our reference-based detector work with a small reference set. In our experiments, we use reference sets of size 10 to 2000. The Mahalanobis distance requires to approximate the covariance matrix of the reference set. In our simulations, the embeddings of dimension 512 make the estimation unreliable. On the contrary, RAINPROOF, which rely on information projections, remains numerically sound with small reference set.

Table 3: Average impact of different OOD detectors on the `BLEU` score for different type of dataset: IN data only, OOD data and the combination of both ALL. For each we report the absolute average `BLEU` score (Abs.), the average gains in `BLEU` (G.s) compared to a setting without OOD filtering ($f_\theta$ only) and the share of the subset removed by the detector (R.Sh.). These results are achieved by setting $\gamma$ such that we remove $20\%$ of the IN dataset.

| | | | IN | | | OOD | | | ALL | | |
|---|---|---|---|---|---|---|---|---|---|---|---|
| | | | Absolute | Gains | Removed shares | Absolute | Gains | Removed shares | Absolute | Gains | Removed shares |
| | | ✗ | 53.6 | +0.0 | 0.0% | 30.8 | +0.0 | 0.0% | 44.4 | +0.0 | 0.0% |
| $s_0$ | Ours | $a_{D_\alpha}$ | 57.2 | +3.6 | 19.7% | 40.9 | +10.1 | 57.1% | 55.6 | +11.2 | 35.1% |
| | | $a_{FR}$ | 56.6 | +3.1 | 19.0% | 39.9 | +9.1 | 60.8% | 54.6 | +10.1 | 36.7% |
| | Bas. | $a_E$ | 56.3 | +2.7 | 20.0% | 31.9 | +1.1 | 31.9% | 48.3 | +3.9 | 24.1% |
| | | $a_L$ | 58.1 | +4.5 | 19.2% | 34.6 | +3.8 | 43.7% | 52.4 | +8.0 | 28.9% |
| | | $a_{MSP}$ | 52.4 | -1.2 | 18.5% | 26.7 | -4.1 | 38.2% | 43.2 | -1.2 | 28.1% |
| $s_1$ | Ours | $a_{D_\alpha^*}$ | 54.0 | +0.4 | 19.2% | 31.6 | +0.8 | 61.3% | 48.9 | +4.5 | 38.0% |
| | | $a_{FR^*}$ | 54.0 | +0.4 | 19.4% | 31.6 | +0.8 | 61.4% | 48.9 | +4.5 | 38.1% |
| | Bas. | $a_C$ | 54.2 | +0.7 | 14.6% | 31.3 | +0.5 | 17.9% | 46.1 | +1.6 | 15.5% |
| | | $a_M$ | 53.6 | +0.1 | 20.0% | 31.6 | +0.9 | 59.0% | 47.4 | +3.0 | 37.9% |

## 5 TOWARDS A PRACTICAL EVALUATION OF OOD DETECTORS

Following previous work, we measure the performance of the detectors on the OOD detection task based on `AUROC` and `FPR`. However, this evaluation framework neglects the impact of the detector on the overall system's performance. We identify three main evaluation criteria that are important in practice: execution time, overall system performance in terms of quality of the generated sentences, and interpretability of the decision. Our study is conducted on NMT because due to the existence of relevant and widely adopted metrics for assessing the quality of a generated sentence (*i.e.*, `BLEU` (Papineni et al., 2002) and BERTSCORE (`BERT-S`) (Unanue et al., 2021)).

### 5.1 COMPLEXITY STUDY

**Runtime and memory costs.** We report in Tab. 1b the runtime of all methods. Detectors for $s_0$ are faster than the ones for $s_1$. Contrarily to detectors using references, the no-reference detectors do not require additional memory. They can be setup easily in a plug&play manner at the output of any model.

**Numerical stability.** The Mahalanobis distance requires to estimate both $\mu$ and $\Sigma^{-1}$ (see Sec. A.6). The dimension of the latent space of the considered pre-trained model is either 768 or 512. In this setting, when the size of the reference set is small, the estimation of the Mahalanobis parameters is numerically unstable. For $s_1$, `RAINPROOF` relies on information projection and does not involve numerically unstable computations but requires a larger memory footprint (0.5 GB) to store the reference set (2000 probability distributions of dimension 50K).

### 5.2 IMPACT OF OOD FILTERING ON TRANSLATION QUALITY

The main objective of OOD filtering is to remove samples that are far from the training distribution. On these samples, the user has no guarantee that the model will produce a good quality translation. In this experiment, we compare the performance of the system with and without the different detectors in terms of the quality of the generated sentence.

**Global performance.** In Tab. 3, we report the global performance of the systems ($f_\theta$) without and with OOD detectors on IN samples, OOD samples and all samples (ALL). From the first row of Tab. 3, we notice that OOD samples are harmful to the model. We observe that, in most of the cases, adding detectors increases the model performance on IN, OOD and all samples. Exceptions include $a_{MSP}$ (for OOD, IN and ALL) and $a_M$ (for OOD). Results indicate that no-reference `RAINPROOF` outperforms the reference-based version of `RAINPROOF`. Thus, **OOD detector evaluation should consider the final task performance.** Overall, it is worth noting that directly adapting classical OOD detection methods (*e.g.*, MSP or Energy) to the sequence generation problem leads to poor results in terms of performance gains (*i.e.*, as measured by `BLEU` or `BERT-S`). **In others words, the final task does not benefit from adding classical OOD detectors.**

**Finer performance analysis.** In Tab. 4, we report the per-shift-types performance of $f_\theta$ with and without OOD detector. In Tab. 4, we observe a decrease in performance in the case of language and

Table 4: Detailed impacts on NMT performance results per tasks (Domain- or Language-shifts) of the different OOD detectors. We present results on the different part of the data: IN data, OOD data and the combination of both, ALL. For each we report the absolute average BLEU score (Abs.), the average gains in BLEU (G.s.) compared to a setting without OOD filtering ($f_\theta$ only) and the share of the subset removed by the detector (R.Sh.). We provide more detailed results on each dataset in Ap. D

| | | | Domain shifts | | | | | | | | Language shifts | | | | | | | | |
| | | | IN | | | OOD | | | ALL | | | IN | | | OOD | | | ALL | | |
| | | | Abs. | G. | Rh. | Abs. | G. | Rh. | Abs. | G. | Rh. | Abs. | G. | Rh. | Abs. | G. | Rh. | Abs. | G. | Rh. |
|---|---|---|---|---|---|---|---|---|---|---|---|---|---|---|---|---|---|---|---|---|
| | | $\mathcal{X}$ | 46.9 | +0.0 | 0.0% | 43.3 | +0.0 | 0.0% | 45.1 | +0.0 | 0.0% | 60.2 | +0.0 | 0.0% | 18.3 | +0.0 | 0.0% | 43.8 | +0.0 | 0.0% |
| $s_0$ | Ours | $a_{D_\alpha}$ | 50.5 | +3.6 | 19.6% | 48.5 | +5.2 | 29.8% | 50.5 | +5.4 | 24.7% | 63.8 | +3.6 | 19.8% | 33.3 | +15.0 | 84.3% | 60.7 | +17.0 | 45.4% |
| | | $a_{FR}$ | 49.7 | +2.8 | 19.3% | 47.6 | +4.4 | 40.6% | 49.2 | +4.1 | 29.9% | 63.5 | +3.3 | 18.7% | 32.1 | +13.8 | 81.0% | 60.0 | +16.2 | 43.5% |
| | Bas. | $a_E$ | 49.3 | +2.4 | 20.0% | 45.5 | +2.2 | 17.8% | 47.6 | +2.5 | 18.9% | 63.3 | +3.1 | 20.0% | 18.4 | +0.1 | 46.0% | 49.1 | +5.3 | 29.4% |
| | | $a_L$ | 50.7 | +3.8 | 19.2% | 47.5 | +4.3 | 24.2% | 49.9 | +4.8 | 21.7% | 65.4 | +5.2 | 19.2% | 21.6 | +3.3 | 63.2% | 54.9 | +11.1 | 36.1% |
| | | $a_{MSP}$ | 45.8 | -1.1 | 19.2% | 33.4 | -9.9 | 45.7% | 40.8 | -4.3 | 32.5% | 59.0 | -1.3 | 17.7% | 20.1 | +1.8 | 30.8% | 45.7 | +1.9 | 23.7% |
| $s_1$ | Ours | $a_{D_\alpha^*}$ | 47.1 | +0.2 | 18.9% | 37.8 | -5.4 | 62.7% | 45.9 | +0.9 | 40.8% | 60.9 | +0.6 | 19.5% | 25.4 | +7.1 | 60.0% | 51.9 | +8.2 | 35.2% |
| | | $a_{FR^*}$ | 47.1 | +0.2 | 18.9% | 37.7 | -5.5 | 62.6% | 46.0 | +0.9 | 40.7% | 60.9 | +0.7 | 19.9% | 25.5 | +7.2 | 60.2% | 52.0 | +8.2 | 35.5% |
| | Bas. | $a_C$ | 47.6 | +0.7 | 13.6% | 43.5 | +0.3 | 3.6% | 45.4 | +0.3 | 8.6% | 60.9 | +0.7 | 15.6% | 19.1 | +0.8 | 32.2% | 46.7 | +2.9 | 22.5% |
| | | $a_M$ | 46.9 | -0.0 | 20.0% | 43.0 | -0.3 | 61.9% | 44.4 | -0.7 | 41.0% | 60.4 | +0.1 | 20.0% | 20.3 | +2.0 | 56.0% | 50.4 | +6.6 | 34.9% |

domain shifts, the latter being more harmful. On domain shifts, we observe that reference-based detectors decrease system's performance on OOD samples. This means that the detectors tend to filter out samples that are well-handled by the model and ignore sentences that are not. It is worth noting that reference-based detectors remove, in proportion, twice as many samples as their no-reference counterparts, while the threshold selection procedure remains the same. This observation also holds when removing less samples (*i.e.*, calibrating $\gamma$ that we remove 10%, 5% or even 1% of the IN dataset) (Tab. 15).

**Threshold free analysis.** In Tab. 2, we report the correlation between OOD scores and final task performance for the case of domain shifts. We refer the reader to Tab. 14 for the results on language shifts. We observe that the likelihood score is the most correlated with the final sentence quality, as measured by BLEU or BERT-S. This finding illustrates that higher correlation with sentence quality does not necessarily translate into higher performance gains when filtering OOD samples. *This result suggests that Quality Estimation* (Specia et al., 2010; Blatz et al., 2004)*, while closely related, is a different problem.*

### 5.3 TOWARDS AN INTERPRETABLE DECISION

An important dimension fostering adoption is the ability to verify the decision taken by the automatic system (Montavon et al., 2018). RAINPROOF offers a step in this direction when used with references: for each input sample, RAINPROOF finds the closest sample (in the sens of the Information Projection) in the reference set to take its decision. We present in Tab. 5 some OOD samples along with their translation scores, projection scores, and their projection on the reference set. We notice that, in general, sentences that are close to the reference set, and whose projection has a close meaning, are better handled by $f_\theta$. Therefore, one can visually interpret the prediction of RAINPROOF, and validate it. This observation further validate our method.

### 6 CONCLUSIONS

In this work, we introduced both a detection framework called RAINPROOF as well as a new benchmark called LOFTER for detecting OOD samples when using textual generators in the black-box scenario. Our work adopts an operational perspective by not only considering OOD performance but also task-specific metrics. Our results show that, despite the good results obtained in

Table 5: OOD inputs, their translations and projections onto the reference set. The first 2 are far from the reference set and not well translated whereas the next 2 are very close to the reference set and well translated. We can, for that matter, notice that the projection is quite close to the input sentence grammatically speaking.

| | | |
|---|---|---|
| Source | Ahir a la nit vàrem treballar fins a les deu. | |
| Ground truth | Last night we worked until 10 p.m. | |
| Generated | Ahir a la nit vàrem treballar fins a les deu. | BLEU 3.75 |
| $\mathbf{p}^*(\mathbf{x})$ | Dar gato por liebre. | Score 1.23 |
| Source | Aquesta cola s'ha esbravat i no té bon gust. | |
| Ground-truth | This cola has lost its fizz and doesn't taste any good. | |
| Generated | This tail s'ha esbravat i no tea bon gust. | BLEU 4.09 |
| $\mathbf{p}^*(\mathbf{x})$ | Esta cuchara es de té. | Score 1.14 |
| source | Aquesta és una carta molt estranya. | |
| Ground-truth | This is a very strange letter. | |
| Generated | This is a molt estranya card. | BLEU 26.27 |
| $\mathbf{p}^*(\mathbf{x})$ | Este carro es chiquito. | Score 0.74 |
| source | Austràlia no és Àustria. | |
| Ground-truth | Australia isn't Austria. | |
| Generated | Austràlia is not Austria. | BLEU 21.86 |
| $\mathbf{p}^*(\mathbf{x})$ | La vida no es fácil. | Score 0.82 |

pure OOD detection, *OOD filtering can harm the performance of the final system, as it is the case for MSP or Mahanalobis.* We found that, RAINPROOF breaks this curse and induces significant gains in translation performance both on OOD samples and in general. In conclusion, this work paves the way to the development of detectors tailored for text generators and calls for a global evaluation when benchmarking future OOD detectors.

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

# 7 APPENDIX

# A  EXPERIMENTAL SETTING

In this section we dive into the details and definitions of our experimental setting. First we present our OOD detection performance metrics (Sec. A.1), then we provide a couple samples for one of the small language shifts (Sec. A.3). We also discuss the choices of pretrained model (Sec. A.5) and how we adapted common OOD detectors to the text generation case (Sec. A.6).

## A.1  ADDITIONNAL DETAILS ON METRICS

OOD Detection is usually an unbalanced binary classification problem where the class of interest is OUT. Let us denote $Z$ the random variable corresponding to actually being out of distribution. We can assess the performance of our OOD detectors focusing on the **False alarm rate** and on the **True detection rate**. The **False alarm rate** or False positive rate (FPR) is the proportion of samples missclassified as OUT. For a score threshold $\gamma$, we have FPR $= \Pr\left(a(\mathbf{x}) > \gamma \mid Z = 0\right)$. The **True detection rate** or True positive rate (TPR) is the proportion of OOD samples that are detected by the method. It is given by TPR $= \Pr\left(a(\mathbf{x}) > \gamma \mid Z = 1\right)$.

In order to evaluate the performance of our methods we will focus and report mainly the `AUROC` and the `FPR`, we provide more detailed metrics and experiments in Sec. A.1.

**Area Under the Receiver Operating Characteristic curve (`AUROC`) Bradley (1997).** The Receiver Operating Characteristic curve is curve obtained by plotting the True positive rate against the False positive rate. The area under this curve is the probability that an in-distribution example $\mathbf{X}_{in}$ has a anomaly score higher than an OOD sample $\mathbf{x}_{out}$: `AUROC`$= \Pr(a(\mathbf{x}_{in}) > a(\mathbf{x}_{out}))$. It is given by $\gamma \mapsto (\Pr\left(a(\mathbf{x}) > \gamma \mid Z = 0\right), \Pr\left(a(\mathbf{x}) > \gamma \mid Z = 1\right))$.

**False Positive Rate at $95\%$ True Positive Rate (`FPR`).** We accept to allow only a given false positive rate $r$ corresponding to a defined level of safety and we want to know what share of positive samples we actually catch under this constraint. It leads to select a threshold $\gamma_r$ such that the corresponding TPR equals $r$. At this threshold, one then computes: $\Pr(a(\mathbf{x}) > \gamma_r \mid Z = 0)$ with $\gamma_r$ s.t. TPR$(\gamma_r) = r$. $r$ is chosen depending of the difficulty of task at hand and the required level of safety.

For the sake of brevity we present only `AUROC` and `FPR` metrics in our aggregated results but we also used Detection error and Area Under the Precision-Recall curve metrics and those are presented in our full results section (Ap. C).

**Detection error**. It is simply the probability of miss-classification for a given True positive rate.

**Area Under the Precision-Recall curve (`AUPR-IN/AUPR-OUT`) Davis & Goadrich (2006).** The Precision-Recall curve plots the recall (true detection rate) against the precision (actual proportion of OOD amongst the predicted OOD). The area under this curve $\gamma \mapsto (\Pr\left(Z = 1 \mid s(\mathbf{X}) \leqslant \gamma\right), \Pr\left(s(\mathbf{X}) \leqslant \gamma \mid Z = 1\right))$ captures the trade-off between precision and recall made by the model. A high value represents a high precision and a high recall *i.e.* the detector captures most of the positive samples while having few False positive.

## A.2  LANGUAGE PAIRS

| Model | IN data | OUT data |
|-------|---------|----------|
| **Language shift** | | |
| DE-EN | Tatoeba DE | News FR |
| DE-EN | Tatoeba DE | Tatoeba NLD |
| ES-EN | Tatoeba ES | News FR |
| ES-EN | Tatoeba ES | Tatoeba CAT |
| ES-EN | Tatoeba ES | Tatoeba POR |
| NLD-EN | Tatoeba ES | AFR |
| **Domain shift** | | |
| DE-EN | Tatoeba DE | EMEA DE |
| DE-EN | Tatoeba DE | Eurparl DE |
| ES-EN | Tatoeba ES | EMEA DE |
| ES-EN | Tatoeba ES | Eurparl DE |

Table 6: Summary of models and studied shifts.

## A.3 SAMPLES

| Source sentence | Expected translation | Translation | BLEU |
|---|---|---|---|
| A en Tom li agrada la tecnologia. | Tom likes technology. | Tom li likes technology. | 42.73 |
| Ací està la teua bossa. | Here is your bag. | Ací está la teua bossa. | 8.12 |
| Això et posarà en perill. | That'll put you in danger. | Això et posarà en perill. | 8.12 |
| A Londres hi han molts parcs bonics. | There are many beautiful parks in London. | To London hi han molts parcs bonics. | 6.57 |
| Aquest pa és molt deliciós. | This bread is very delicious. | Aquest pa és molt deliciós. | 8.12 |
| A tots els meus amics els agraden els videojocs. | All my friends like playing videogames. | A tots els meus amics els agrade els videojocs. | 4.20 |
| Açò és un peix. | This is a fish. | Aaaaaaaaaaaaaaaaaaaa ... aaaaaaaaaaaaaaaaaaaaaaaaaaa | 0.00 |
| Moltes felicitats! | Congratulations! | Moltes congrats! | 27.52 |
| Bon any nou! | Happy New Year! | Bon any nou! | 15.97 |
| Aquell que menteix, robarà. | He that will lie, will steal. | The one who's mindless, he'll steal. | 12.22 |
| Jo sóc qui té la clau. | I'm the one who has the key. | Jo soc qui te la clau. | 5.69 |
| En Tom surt a treballar cada matí a dos quarts de set. | Tom leaves for work at 6:30 every morning. | In Tom surt to pull each matí to two quarts of set. | 3.67 |
| Ell m'ha dit que la seva casa era embruixada. | He told me that his house was haunted. | Ell m'ha dit that the seva house was haunted. | 27.78 |
| Aquest és el lloc on va nèixer el meu pare. | This is the place where my father was born. | Aquest is the lloc on va nèixer el meu pare. | 8.30 |

Table 7: Example of behavior of a language model trained to handle Spanish inputs on Catalan inputs.

## A.4 DIALOG DATASETS

**Switchboard Dialog Act Corpus (`SwDA`)** is a corpus of telephonic conversations. The corpus provides labels, topic and speaker information (Stolcke et al., 2000).

**ICSI MRDA Corpus (`MRDA`)** contains transcript 75h of naturally occuring meetings involving more than 50 people (Shriberg et al., 2004).

**DaylyDialog Act Corpus (`DyDA`)** contains daily common communications between people, covering topic such as small talk, meteo or daily activities (Li et al., 2017).

**Interactive Emotional Dyadic Motion Capture IEMOCAP)**(Tripathi et al., 2018) consists of transcripts of improvisations or scripted scenarii supposed to outline the expression of emotions.

## A.5 CHOICES OF MODELS

A lot of pretrained model for conditional text generation are available. To perform our experiments we needed models that were already well installed and deployed and that would also support OOD settings. For translation tasks we needed specialized models for a notion of OOD to be easily defined. It would be indeed more hazardous to define a notion of OOD language when working with a multilingual model. The same is true for conversational models.

**Neural Machine Translation model.** We benchmark our OOD method on translation models provided by Helsinky NLP Tiedemann & Thottingal (2020) on several pairs of languages with large and small shifts. We extended the experiment to detect domain shifts. These models are indeed specialized in each language pairs and are widely recognize in the neural machine translation field. For our experiments we used the testing set provided along these models, so we can consider that they have been fine tuned over the same distribution.

**Conversational model.** We used a dialogGPT Zhang et al. (2019) model fine-tuned on the `Multi WOZ` dataset as chat bot model. The finetuning on daily dialogue type tasks ensure that the model is specialized, thus allowing us to get a good definition of samples not being in its range of expertise. Moreover, the choice of the architure, DialogGPT, guarantee that our results are valid on a very common architecture.

**Additional finetuning.** We further finetuned the models on the reference set to check whether additional finetuning on the distribution would affect the results. It did not change significantly the results Tab. 17. It is not surprising considering that the models we used were already trained on a very similar distribution.

## A.6 GENERALIZATION OF EXISTING OOD DETECTORS TO SEQUENCE GENERATION

In this section, we extend classical OOD detection score to the conditional text generation settting. Common OOD detectors were built for classification tasks and we need to adapt them to conditional text generation. Our task can be viewed as a sequence of classification problems with a very large number of classes (the size of the vocabulary). We chose the most naive approach which consists of

averaging the OOD scores over the sequence. We experimented with other aggregation such as the min/max or the standard deviation without getting interesting results.

**Likelihood Score** The most naive approach to build a OOD score is to rely solely on the log-likelihood of the sequence. For a conditioning $\mathbf{x}$ we define the log-likelyhood score by $a_L(\mathbf{x}) = -\sum_{t=0}^{|\hat{\mathbf{y}}|-1} \log \mathbf{p}_\theta(\hat{\mathbf{y}}_{t+1}|\mathbf{x}, \hat{\mathbf{y}}_{\leqslant t})$. The likelihood is the same as the perplexity.

**Average Maximum Softmax Probability score** The maximum softmax probability Hendrycks & Gimpel (2017) takes the probability of the mode of the categorical distribution as score of OOD. We extend thise definition in the case of sequence of probability distribution by averaging this score along the sequence. For a given conditioning $\mathbf{x}$, we define the average MSP score $a_{\text{MSP}}(\mathbf{x}) = \frac{1}{|\hat{\mathbf{y}}|} \sum_{t=1}^{|\hat{\mathbf{y}}|} \max_{i \in [|0,K|]} \mathbf{p}_\theta^T(i|\mathbf{x}, \hat{\mathbf{y}}_{\leqslant t}))$. While it is closely linked to uncertainty measures it discards most of the information contained in the probability distribution. It discards the whole probability distribution. We claim that much more information can be retrieve by studying the whole distribution.

**Average Energy score** We extend the definition of the energy score described in Liu et al. (2020) to a sequence of probability distributions by averaging the score along the sequence. For a given conditioning $\mathbf{x}$ and a temperature $T$ we define the average energy of the sequence: $a_E(\mathbf{x}) \triangleq -\frac{T}{|\hat{\mathbf{y}}|} \sum_{t=1}^{|\hat{\mathbf{y}}|} \log \sum_{i}^{|\Omega|} e^{f_\theta(\mathbf{x}, \hat{\mathbf{y}}_{\leqslant t})_i/T}$. It corresponds to the normalization term of the softmax function applied on the logits. While it takes into account the whole distribution, it only takes into account the amount of unormalized mass before normalization without attention to how this mass is distributed along the features.

**Mahalanobis distance** Following Lee et al. (2018c) compute the Mahalanobis matrice based on the samples of a given reference set $\mathcal{R}$. In our case we are using encoder-decoder models we use the output of the last hidden layer of the encoder as embedding. Let's denote $\phi(\mathbf{x})$ this embedding for a conditionning $\mathbf{x}$. Let's $\mu$ and $\Sigma$ be respectively the mean and the covariance of these embedding on the reference set. We define $a_M(\mathbf{x}) = \left(1 + (\phi(\mathbf{x}) - \mu)^\top \Sigma^{-1}(\phi(\mathbf{x}) - \mu)\right)^{-1}$.

# B    PARAMETERS TUNING

Detectors depend on their anomaly score to make decision and these scores can be parametric. First of all, soft probability based scores depend on the soft probability distribution and its scaling, therefore the temperature is a crucial parameter to tune to get the most performance. While a small temperature tend to make the distribution more picky, higher value spread the probability mass along the classes. Moreover, the renyi divergence and its related informaiton projection depend on a factor $\alpha$. We provide here further results and analysis of those parameters on our results.

## B.1    IMPACT OF $\alpha$

Indeed, in Fig. 1 we present the impact of the size of the reference set and of the paramet $\alpha$ on Renyi information projection to distinguish dialog shifts, as expected, the larger the reference set, the better. However, we see that smaller values of $\alpha$ yield better results.

We recall that the Renyi divergence is defined as $D_\alpha(\mathbf{p}\|\mathbf{q}) = \frac{1}{\alpha-1} \log \left(\sum_{i=1}^{|\Omega|} \frac{p_i^\alpha}{q_i^{\alpha-1}}\right)$, where $\alpha \in \mathbb{R}_+ - \{1\}$. Smaller values of $\alpha$ distribute the weight of each feature more equally in the final divergence, more specifically they tend to give an equal weight to the very likely outcome as well as to the less likely ones, therefore giving more weight to the tail of the distribution. When $\alpha$ tends to $0$ the Renyi divergence actually counts the number of nonzero common probabilities. That makes sens in terms of topic detection, it counts the common tokens considered during text generation.

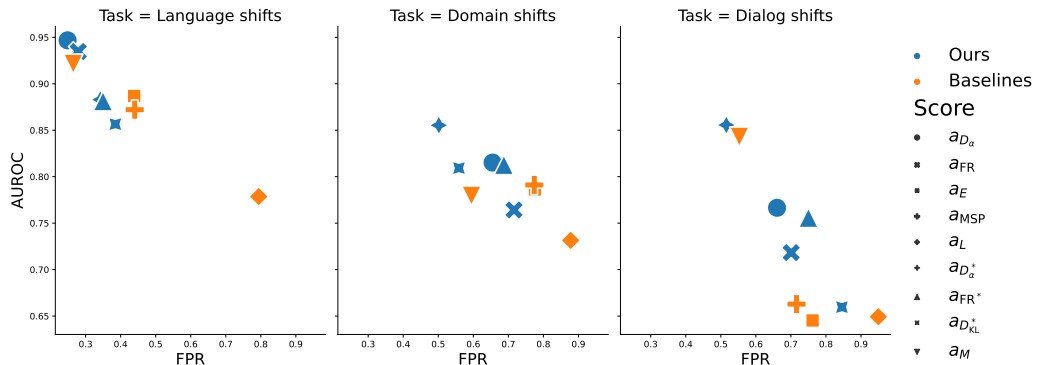

Figure 2: Trade-offs between AUROC and FPR for each tasks and metrics

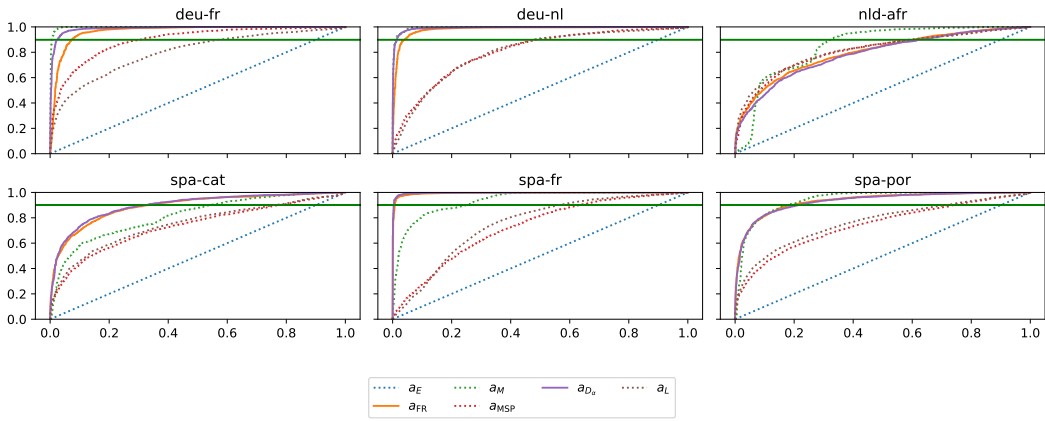

Figure 3: ROCAUC curves for our uncertainty based metrics compared to common baselines for language shifts detection. Baselines are represented in dashed lines.

# C  PERFORMANCE OF OUR DETECTORS IN OOD DETECTION

## C.1  SUMMARY OF OUR RESULTS

In Fig. 2 we present the different level of performance of all the detectors we studied. We can see that in every task our detectors outperform the baselines but also that in dialog shift, while the Mahalanobis distance outperform clearly our detectors for $\mathfrak{s}_0$, they still outperform baselines for their scenario by far.

## C.2  DETAILED RESULTS OF OOD DETECTION PERFORMANCES

In this section we present the performances of our OOD detectors on each detailed tasks, *i.e.* for each pair of IN and OOD data with all the considered metrics. We show that our metrics outperform other OOD detectors baselines in almost all scenarios.

Table 8: Detailed results of the performances of our OOD detectors on different language shifts. The first language of the pair is the reference language of the model and the second one is the studied shift.

| Scenario | | Score | AUPR-IN | AUPR-OUT | AUROC | ERR | f1 | FPR | precision | recall |
|---|---|---|---|---|---|---|---|---|---|---|
| s0 | Ours | $a_{D_\alpha}$ | **1.00** | **0.99** | **1.00** | **0.03** | **0.91** | **0.01** | **0.83** | **1.00** |
| | | $a_{FR}$ | 0.98 | **0.99** | 0.99 | 0.05 | **0.91** | 0.05 | **0.83** | 0.99 |
| | Baselines | $a_E$ | 0.98 | 0.95 | 0.98 | 0.04 | 0.90 | 0.04 | **0.83** | **1.00** |
| | | $a_L$ | 0.80 | 0.71 | 0.78 | 0.44 | 0.69 | 0.84 | 0.75 | 0.63 |
| | | $a_{MSP}$ | 0.99 | 0.98 | 0.99 | 0.04 | 0.90 | 0.03 | **0.83** | **1.00** |
| s1 | Ours | $a_{D_z^*}$ | **1.00** | **1.00** | **1.00** | 0.03 | **0.91** | **0.00** | **0.83** | **1.00** |
| | | $a_{D_{KL}^*}$ | 0.99 | **1.00** | **1.00** | 0.03 | **0.91** | 0.01 | **0.83** | **1.00** |
| | | $a_{D_g^{\text{diff}}}$ | 0.99 | 0.98 | 0.99 | 0.06 | 0.90 | 0.06 | **0.83** | 0.99 |
| | | $a_{D_{KL}^{\text{diff}}}$ | 0.99 | 0.98 | 0.98 | 0.06 | 0.90 | 0.07 | **0.83** | 0.98 |
| | | $a_{D^{\text{mean}}}$ | **1.00** | **1.00** | **1.00** | **0.02** | **0.91** | **0.00** | **0.83** | **1.00** |
| | | $a_{D_{KL}^{\text{mean}}}$ | **1.00** | 0.99 | **1.00** | **0.02** | **0.91** | **0.00** | **0.83** | 0.99 |
| | | $a_{FR^*}$ | **1.00** | **1.00** | **1.00** | 0.03 | **0.91** | **0.00** | **0.83** | **1.00** |
| | | $a_{FR^{\text{diff}}}$ | 0.57 | 0.58 | 0.59 | 0.47 | 0.38 | 0.90 | 0.59 | 0.28 |
| | | $a_{FR^{\text{mean}}}$ | 0.99 | 0.97 | 0.99 | 0.03 | 0.90 | **0.00** | **0.83** | 0.99 |
| | Baselines | $a_C$ | 0.76 | 0.82 | 0.81 | 0.34 | 0.42 | 0.64 | 0.56 | 0.34 |
| | | $a_M$ | 0.79 | 0.86 | 0.82 | 0.25 | 0.68 | 0.44 | 0.75 | 0.62 |

(a) deu-fr

| Scenario | | Score | AUPR-IN | AUPR-OUT | AUROC | ERR | f1 | FPR | precision | recall |
|---|---|---|---|---|---|---|---|---|---|---|
| s0 | Ours | $a_{D_\alpha}$ | 0.94 | **0.94** | 0.94 | 0.17 | **0.87** | 0.29 | **0.82** | **1.00** |
| | | $a_{FR}$ | **0.95** | **0.94** | **0.95** | **0.16** | **0.87** | **0.28** | **0.82** | 0.92 |
| | Baselines | $a_E$ | 0.82 | 0.75 | 0.79 | 0.45 | 0.71 | 0.85 | 0.77 | **1.00** |
| | | $a_L$ | 0.80 | 0.71 | 0.77 | 0.46 | 0.67 | 0.88 | 0.50 | **1.00** |
| | | $a_{MSP}$ | 0.77 | 0.76 | 0.74 | 0.45 | 0.67 | 0.85 | 0.70 | **1.00** |
| s1 | Ours | $a_{D_z^*}$ | 0.81 | 0.72 | 0.77 | 0.44 | 0.70 | 0.84 | 0.76 | 0.65 |
| | | $a_{D_{KL}^*}$ | 0.77 | 0.70 | 0.75 | 0.46 | 0.68 | 0.88 | 0.75 | 0.62 |
| | | $a_{D_g^{\text{diff}}}$ | 0.80 | 0.74 | 0.78 | 0.43 | 0.69 | 0.81 | 0.76 | 0.64 |
| | | $a_{D_{KL}^{\text{diff}}}$ | 0.71 | 0.59 | 0.67 | 0.50 | 0.58 | 0.95 | 0.71 | 0.50 |
| | | $a_{D^{\text{mean}}}$ | **0.86** | 0.77 | **0.83** | 0.42 | **0.76** | 0.80 | 0.78 | 0.73 |
| | | $a_{D_{KL}^{\text{mean}}}$ | 0.77 | 0.58 | 0.70 | 0.52 | 0.66 | 0.99 | 0.75 | 0.59 |
| | | $a_{FR^*}$ | 0.79 | 0.72 | 0.77 | 0.45 | 0.69 | 0.84 | 0.75 | 0.63 |
| | | $a_{FR^{\text{diff}}}$ | 0.51 | 0.51 | 0.51 | 0.50 | 0.31 | 0.95 | 0.52 | 0.22 |
| | | $a_{FR^{\text{mean}}}$ | 0.60 | 0.44 | 0.48 | 0.52 | 0.00 | 1.00 | 0.00 | 0.00 |
| | Baselines | $a_C$ | 0.67 | 0.67 | 0.70 | 0.44 | 0.63 | 0.85 | 0.71 | 0.60 |
| | | $a_M$ | 0.74 | **0.84** | 0.82 | **0.30** | **0.76** | 0.56 | **0.79** | **0.74** |

(b) spa-por

| Scenario | | Score | AUPR-IN | AUPR-OUT | AUROC | ERR | f1 | FPR | precision | recall |
|---|---|---|---|---|---|---|---|---|---|---|
| s0 | Ours | $a_{D_\alpha}$ | **0.86** | **0.94** | **0.91** | 0.29 | **0.77** | 0.42 | **0.70** | **1.00** |
| | | $a_{FR}$ | **0.86** | 0.93 | 0.90 | 0.32 | **0.77** | 0.47 | **0.70** | 0.86 |
| | Baselines | $a_E$ | 0.70 | 0.82 | 0.77 | 0.59 | 0.63 | 0.88 | 0.63 | **1.00** |
| | | $a_L$ | 0.68 | 0.79 | 0.75 | 0.59 | 0.52 | 0.88 | 0.35 | **1.00** |
| | | $a_{MSP}$ | 0.68 | 0.83 | 0.73 | 0.59 | 0.52 | 0.88 | 0.52 | **1.00** |
| s1 | Ours | $a_{D_z^*}$ | 0.58 | 0.78 | 0.69 | 0.60 | 0.54 | 0.89 | 0.58 | 0.51 |
| | | $a_{D_{KL}^*}$ | 0.52 | 0.74 | 0.66 | 0.61 | 0.49 | 0.92 | 0.54 | 0.45 |
| | | $a_{D_g^{\text{diff}}}$ | 0.59 | 0.77 | 0.70 | 0.59 | 0.54 | 0.89 | 0.58 | 0.50 |
| | | $a_{D_{KL}^{\text{diff}}}$ | 0.48 | 0.69 | 0.60 | 0.64 | 0.43 | 0.96 | 0.50 | 0.37 |
| | | $a_{D^{\text{mean}}}$ | **0.68** | 0.80 | **0.76** | 0.60 | **0.62** | 0.90 | **0.62** | 0.61 |
| | | $a_{D_{KL}^{\text{mean}}}$ | 0.56 | 0.67 | 0.62 | 0.66 | 0.51 | 0.99 | 0.56 | 0.47 |
| | | $a_{FR^*}$ | 0.55 | 0.76 | 0.68 | 0.60 | 0.52 | 0.89 | 0.56 | 0.49 |
| | | $a_{FR^{\text{diff}}}$ | 0.35 | 0.65 | 0.50 | 0.63 | 0.27 | 0.95 | 0.37 | 0.21 |
| | | $a_{FR^{\text{mean}}}$ | 0.51 | 0.63 | 0.55 | 0.66 | 0.00 | 0.99 | 0.00 | 0.00 |
| | Baselines | $a_C$ | 0.55 | 0.77 | 0.67 | 0.57 | 0.53 | 0.85 | 0.55 | 0.57 |
| | | $a_M$ | 0.52 | **0.82** | 0.71 | **0.53** | 0.53 | **0.79** | 0.57 | 0.49 |

(c) spa-cat

| Scenario | | Score | AUPR-IN | AUPR-OUT | AUROC | ERR | f1 | FPR | precision | recall |
|---|---|---|---|---|---|---|---|---|---|---|
| s0 | Ours | $a_{D_\alpha}$ | **1.00** | **1.00** | **1.00** | 0.03 | **0.91** | 0.01 | **0.83** | **1.00** |
| | | $a_{FR}$ | **1.00** | **1.00** | **1.00** | 0.03 | **0.91** | 0.01 | **0.83** | **1.00** |
| | Baselines | $a_E$ | 0.99 | 0.96 | 0.98 | 0.04 | 0.90 | 0.03 | **0.83** | **1.00** |
| | | $a_L$ | 0.69 | 0.72 | 0.73 | 0.41 | 0.60 | 0.76 | 0.71 | 0.52 |
| | | $a_{MSP}$ | 0.98 | 0.98 | 0.98 | 0.05 | 0.90 | 0.04 | **0.83** | **1.00** |
| s1 | Ours | $a_{D_z^*}$ | **1.00** | 0.99 | 0.99 | 0.03 | **0.91** | 0.01 | **0.83** | **1.00** |
| | | $a_{D_{KL}^*}$ | 0.94 | 0.97 | 0.96 | 0.07 | 0.90 | 0.09 | **0.83** | 0.99 |
| | | $a_{D_g^{\text{diff}}}$ | 0.97 | 0.98 | 0.98 | 0.08 | 0.90 | 0.11 | **0.83** | 0.98 |
| | | $a_{D_{KL}^{\text{diff}}}$ | 0.97 | 0.97 | 0.97 | 0.09 | 0.90 | 0.14 | **0.83** | 0.97 |
| | | $a_{D^{\text{mean}}}$ | **1.00** | **1.00** | **1.00** | 0.02 | **0.91** | 0.00 | **0.83** | **1.00** |
| | | $a_{D_{KL}^{\text{mean}}}$ | **1.00** | 0.99 | 0.99 | 0.03 | 0.90 | 0.01 | **0.83** | 0.99 |
| | | $a_{FR^*}$ | 0.99 | 0.99 | 0.99 | 0.03 | 0.90 | 0.02 | **0.83** | **1.00** |
| | | $a_{FR^{\text{diff}}}$ | 0.57 | 0.56 | 0.58 | 0.48 | 0.39 | 0.92 | 0.59 | 0.29 |
| | | $a_{FR^{\text{mean}}}$ | 0.99 | 0.97 | 0.99 | 0.04 | 0.90 | 0.02 | **0.83** | 0.98 |
| | Baselines | $a_C$ | 0.67 | 0.72 | 0.72 | 0.43 | 0.00 | 0.82 | 0.00 | 0.00 |
| | | $a_M$ | 0.88 | 0.95 | 0.92 | 0.12 | 0.89 | 0.19 | **0.83** | 0.96 |

(d) spa-fr

| Scenario | | Score | AUPR-IN | AUPR-OUT | AUROC | ERR | f1 | FPR | precision | recall |
|---|---|---|---|---|---|---|---|---|---|---|
| s0 | Ours | $a_{D_\alpha}$ | **0.76** | **0.90** | **0.85** | 0.51 | 0.65 | **0.72** | 0.61 | **1.00** |
| | | $a_{FR}$ | 0.71 | 0.88 | 0.80 | 0.53 | 0.63 | 0.75 | 0.60 | 0.66 |
| | Baselines | $a_E$ | 0.74 | 0.88 | 0.82 | 0.55 | **0.67** | 0.78 | **0.62** | **1.00** |
| | | $a_L$ | 0.72 | 0.88 | 0.82 | 0.58 | 0.48 | 0.82 | 0.31 | **1.00** |
| | | $a_{MSP}$ | 0.70 | 0.88 | 0.81 | 0.55 | 0.48 | 0.78 | 0.31 | **1.00** |
| s1 | Ours | $a_{D_z^*}$ | 0.71 | 0.94 | **0.87** | **0.22** | 0.69 | **0.30** | 0.64 | 0.75 |
| | | $a_{D_{KL}^*}$ | 0.66 | 0.94 | 0.86 | 0.23 | 0.65 | 0.32 | 0.61 | 0.69 |
| | | $a_{D_g^{\text{diff}}}$ | 0.41 | 0.74 | 0.59 | 0.64 | 0.37 | 0.91 | 0.43 | 0.32 |
| | | $a_{D_{KL}^{\text{diff}}}$ | 0.34 | 0.70 | 0.53 | 0.67 | 0.00 | 0.96 | 0.00 | 0.00 |
| | | $a_{D^{\text{mean}}}$ | **0.75** | 0.88 | 0.83 | 0.57 | 0.46 | 0.81 | 0.50 | 0.43 |
| | | $a_{D_{KL}^{\text{mean}}}$ | 0.43 | 0.70 | 0.56 | 0.68 | 0.00 | 0.96 | 0.00 | 0.00 |
| | | $a_{FR^*}$ | 0.70 | **0.95** | **0.87** | **0.22** | 0.67 | **0.30** | 0.62 | 0.72 |
| | | $a_{FR^{\text{diff}}}$ | 0.34 | 0.71 | 0.53 | 0.65 | 0.00 | 0.93 | 0.00 | 0.00 |
| | | $a_{FR^{\text{mean}}}$ | 0.65 | 0.84 | 0.76 | 0.61 | 0.00 | 0.87 | 0.00 | 0.00 |
| | Baselines | $a_C$ | 0.44 | 0.79 | 0.65 | 0.62 | 0.40 | 0.89 | 0.40 | 0.40 |
| | | $a_M$ | 0.47 | 0.85 | 0.71 | 0.55 | 0.49 | 0.78 | 0.52 | 0.47 |

(e) nld-afr

| Scenario | | Score | AUPR-IN | AUPR-OUT | AUROC | ERR | f1 | FPR | precision | recall |
|---|---|---|---|---|---|---|---|---|---|---|
| s0 | Ours | $a_{D_\alpha}$ | **1.00** | 0.99 | 0.99 | 0.03 | **0.91** | 0.01 | **0.83** | **1.00** |
| | | $a_{FR}$ | 0.99 | **0.99** | **0.99** | 0.04 | **0.91** | 0.03 | **0.83** | **1.00** |
| | Baselines | $a_E$ | 0.98 | 0.93 | 0.97 | 0.05 | 0.90 | 0.06 | **0.83** | **1.00** |
| | | $a_L$ | 0.76 | 0.74 | 0.78 | 0.40 | 0.70 | 0.75 | 0.75 | 0.65 |
| | | $a_{MSP}$ | 0.98 | 0.97 | 0.98 | 0.05 | 0.90 | 0.04 | **0.83** | **1.00** |
| s1 | Ours | $a_{D_z^*}$ | **1.00** | **1.00** | **1.00** | 0.03 | **0.91** | **0.00** | **0.83** | **1.00** |
| | | $a_{D_{KL}^*}$ | 0.98 | 0.99 | 0.99 | 0.03 | 0.90 | 0.02 | **0.83** | 0.99 |
| | | $a_{D_g^{\text{diff}}}$ | 0.98 | 0.98 | 0.98 | 0.06 | 0.90 | 0.07 | **0.83** | 0.98 |
| | | $a_{D_{KL}^{\text{diff}}}$ | 0.98 | 0.96 | 0.97 | 0.07 | 0.90 | 0.10 | **0.83** | 0.97 |
| | | $a_{D^{\text{mean}}}$ | **1.00** | 0.99 | **1.00** | **0.02** | **0.91** | **0.00** | **0.83** | 0.99 |
| | | $a_{D_{KL}^{\text{mean}}}$ | 0.99 | 0.97 | 0.99 | 0.03 | 0.90 | **0.00** | **0.83** | 0.99 |
| | | $a_{FR^*}$ | **1.00** | **1.00** | **1.00** | 0.03 | **0.91** | **0.00** | **0.83** | **1.00** |
| | | $a_{FR^{\text{diff}}}$ | 0.57 | 0.57 | 0.58 | 0.48 | 0.38 | 0.92 | 0.58 | 0.28 |
| | | $a_{FR^{\text{mean}}}$ | 0.99 | 0.94 | 0.98 | 0.03 | 0.90 | 0.02 | **0.83** | 0.98 |
| | Baselines | $a_C$ | 0.71 | 0.74 | 0.75 | 0.41 | 0.49 | 0.78 | 0.61 | 0.41 |
| | | $a_M$ | 0.76 | 0.84 | 0.80 | 0.27 | 0.65 | 0.50 | 0.74 | 0.58 |

(f) deu-nl

## C.3 ROC AUC CURVES

### C.3.1 LANGUAGE SHIFTS

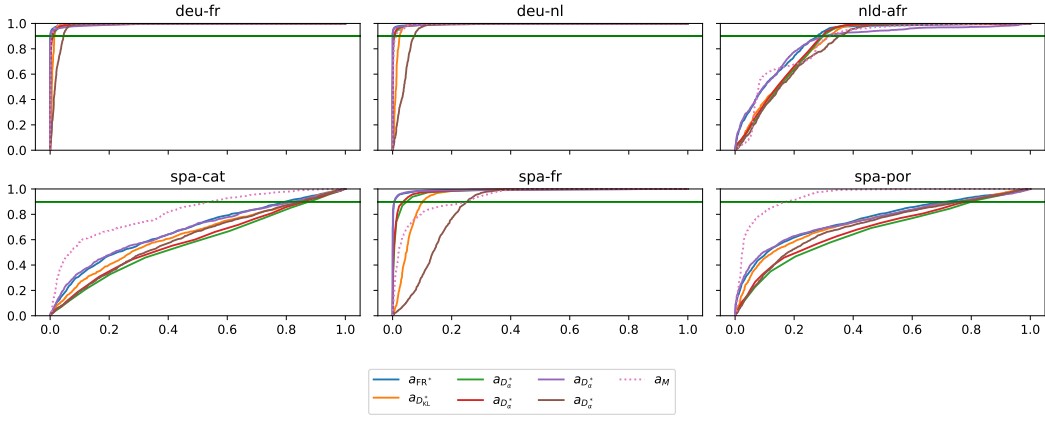

Figure 4: ROC-AUC curves for our reference based metrics compared to common baselines for language shifts detection. Baselines are represented in dashed lines.

Table 9: Detailed results of the performances of our OOD detectors on different domain shifts. For Spanish (spa) and German (de), we present two domains shifts: Technical medical (EMEA) data and legal parlementary texts (parl) against common language embodied by the Tatoeba dataset (tat).

| Scenario | | Score | AUPR-IN | AUPR-OUT | AUROC | ERR | f1 | FPR | precision | recall |
|---|---|---|---|---|---|---|---|---|---|---|
| $s_0$ | Ours | $a_{D_\alpha}$ | **0.90** | **0.76** | **0.86** | **0.43** | 0.81 | **0.82** | 0.80 | 1.00 |
| | | $a_{FR}$ | 0.87 | 0.73 | 0.81 | 0.46 | 0.77 | 0.86 | 0.79 | 0.75 |
| | Baselines | $a_E$ | 0.88 | 0.75 | 0.83 | 0.49 | 0.78 | 0.93 | 0.79 | 1.00 |
| | | $a_L$ | 0.86 | 0.73 | 0.82 | 0.48 | 0.76 | 0.91 | 0.77 | 0.74 |
| | | $a_{MSP}$ | 0.89 | **0.76** | 0.85 | 0.44 | 0.79 | 0.84 | 0.80 | 1.00 |
| $s_1$ | Ours | $a_{D_\alpha^*}$ | **0.90** | 0.88 | **0.90** | 0.25 | 0.82 | 0.45 | 0.81 | 0.86 |
| | | $a_{D_{KL}^*}$ | 0.89 | 0.83 | 0.88 | 0.33 | 0.82 | 0.62 | 0.80 | 0.83 |
| | | $a_{D_\alpha^{diff}}$ | 0.79 | 0.68 | 0.76 | 0.46 | 0.68 | 0.88 | 0.75 | 0.61 |
| | | $a_{D_{KL}^{diff}}$ | 0.79 | 0.68 | 0.75 | 0.47 | 0.66 | 0.89 | 0.75 | 0.59 |
| | | $a_{D_\alpha^{mean}}$ | 0.88 | 0.73 | 0.84 | 0.47 | 0.78 | 0.89 | 0.79 | 0.77 |
| | | $a_{D_{KL}^{mean}}$ | 0.87 | 0.73 | 0.83 | 0.48 | 0.77 | 0.91 | 0.79 | 0.75 |
| | | $a_{FR^*}$ | 0.85 | 0.70 | 0.80 | 0.50 | 0.74 | 0.94 | 0.77 | 0.72 |
| | | $a_{FR^{diff}}$ | 0.52 | 0.53 | 0.53 | 0.49 | 0.32 | 0.93 | 0.53 | 0.22 |
| | | $a_{FR^{mean}}$ | 0.86 | 0.67 | 0.80 | 0.52 | 0.75 | 0.98 | 0.78 | 0.72 |
| | Baselines | $a_C$ | 0.88 | 0.89 | **0.90** | 0.25 | 0.00 | 0.46 | 0.00 | 0.00 |
| | | $a_M$ | 0.87 | **0.90** | 0.89 | **0.22** | 0.81 | **0.39** | 0.80 | 0.81 |

(a) de:news-EMEA

| Scenario | | Score | AUPR-IN | AUPR-OUT | AUROC | ERR | f1 | FPR | precision | recall |
|---|---|---|---|---|---|---|---|---|---|---|
| $s_0$ | Ours | $a_{D_\alpha}$ | **0.75** | **0.75** | **0.76** | **0.41** | 0.67 | **0.76** | 0.67 | 1.00 |
| | | $a_{FR}$ | 0.66 | 0.67 | 0.70 | 0.44 | 0.45 | 0.84 | 0.64 | 0.35 |
| | Baselines | $a_E$ | **0.75** | **0.75** | 0.71 | 0.44 | 0.67 | 0.83 | 0.69 | 1.00 |
| | | $a_L$ | 0.68 | 0.61 | 0.68 | 0.49 | 0.67 | 0.94 | 0.50 | 1.00 |
| | | $a_{MSP}$ | **0.75** | **0.75** | 0.71 | 0.45 | 0.67 | 0.86 | 0.50 | 1.00 |
| $s_1$ | Ours | $a_{D_\alpha^*}$ | 0.66 | 0.65 | 0.68 | 0.44 | 0.00 | 0.84 | 0.83 | 0.00 |
| | | $a_{D_{KL}^*}$ | **0.67** | 0.63 | 0.67 | 0.48 | 0.00 | 0.90 | 0.00 | 0.00 |
| | | $a_{D_\alpha^{diff}}$ | 0.60 | 0.59 | 0.61 | 0.48 | 0.15 | 0.90 | 0.83 | 0.08 |
| | | $a_{D_{KL}^{diff}}$ | 0.60 | 0.59 | 0.61 | 0.48 | 0.00 | 0.91 | 0.00 | 0.00 |
| | | $a_{D_\alpha^{mean}}$ | **0.67** | **0.67** | **0.70** | 0.44 | 0.00 | 0.82 | 0.00 | 0.00 |
| | | $a_{D_{KL}^{mean}}$ | 0.66 | 0.65 | 0.69 | 0.45 | 0.00 | 0.85 | 0.00 | 0.00 |
| | | $a_{FR^*}$ | 0.62 | 0.64 | 0.65 | 0.45 | 0.00 | 0.85 | 0.00 | 0.00 |
| | | $a_{FR^{diff}}$ | 0.51 | 0.52 | 0.51 | 0.49 | 0.00 | 0.94 | 0.00 | 0.00 |
| | | $a_{FR^{mean}}$ | 0.65 | **0.67** | 0.69 | **0.43** | 0.00 | **0.80** | 0.00 | 0.00 |
| | Baselines | $a_C$ | 0.57 | 0.61 | 0.60 | 0.46 | 0.27 | 0.87 | 0.47 | 0.19 |
| | | $a_M$ | 0.62 | 0.66 | 0.66 | 0.44 | 0.00 | 0.83 | 0.00 | 0.00 |

(b) es:news-parl

| Scenario | | Score | AUPR-IN | AUPR-OUT | AUROC | ERR | f1 | FPR | precision | recall |
|---|---|---|---|---|---|---|---|---|---|---|
| $s_0$ | Ours | $a_{D_\alpha}$ | **0.75** | **0.75** | **0.71** | **0.41** | 0.67 | **0.78** | 0.66 | 1.00 |
| | | $a_{FR}$ | 0.61 | 0.65 | 0.65 | 0.45 | 0.42 | 0.84 | 0.61 | 0.32 |
| | Baselines | $a_E$ | **0.75** | **0.75** | 0.68 | 0.45 | 0.67 | 0.85 | 0.66 | 1.00 |
| | | $a_L$ | 0.63 | 0.58 | 0.64 | 0.51 | 0.67 | 0.96 | 0.50 | 1.00 |
| | | $a_{MSP}$ | **0.75** | **0.75** | 0.68 | 0.46 | 0.67 | 0.86 | 0.51 | 1.00 |
| $s_1$ | Ours | $a_{D_\alpha^*}$ | **0.69** | **0.66** | **0.68** | 0.43 | 0.30 | 0.81 | 0.80 | 0.22 |
| | | $a_{D_{KL}^*}$ | 0.66 | 0.64 | **0.68** | 0.46 | 0.00 | 0.88 | 0.00 | 0.00 |
| | | $a_{D_\alpha^{diff}}$ | 0.61 | 0.59 | 0.62 | 0.48 | 0.28 | 0.90 | 0.51 | 0.20 |
| | | $a_{D_{KL}^{diff}}$ | 0.59 | 0.58 | 0.60 | 0.48 | 0.00 | 0.90 | 0.00 | 0.00 |
| | | $a_{D_\alpha^{mean}}$ | 0.65 | 0.65 | **0.68** | 0.45 | 0.00 | 0.86 | 0.00 | 0.00 |
| | | $a_{D_{KL}^{mean}}$ | 0.65 | 0.65 | 0.66 | 0.45 | 0.00 | 0.86 | 0.00 | 0.00 |
| | | $a_{FR^*}$ | 0.63 | 0.64 | 0.66 | 0.45 | 0.00 | 0.86 | 0.00 | 0.00 |
| | | $a_{FR^{diff}}$ | 0.51 | 0.52 | 0.51 | 0.49 | 0.00 | 0.93 | 0.00 | 0.00 |
| | | $a_{FR^{mean}}$ | 0.64 | 0.64 | 0.67 | 0.45 | 0.00 | 0.86 | 0.00 | 0.00 |
| | Baselines | $a_C$ | 0.52 | **0.66** | 0.59 | **0.41** | 0.40 | **0.78** | 0.52 | 0.33 |
| | | $a_M$ | 0.58 | 0.61 | 0.62 | 0.47 | 0.00 | 0.89 | 0.00 | 0.00 |

(c) de:news-parl

| Scenario | | Score | AUPR-IN | AUPR-OUT | AUROC | ERR | f1 | FPR | precision | recall |
|---|---|---|---|---|---|---|---|---|---|---|
| $s_0$ | Ours | $a_{D_\alpha}$ | **0.92** | **0.81** | **0.89** | **0.37** | 0.83 | **0.70** | 0.81 | 1.00 |
| | | $a_{FR}$ | 0.89 | 0.75 | 0.85 | 0.44 | 0.79 | 0.82 | 0.80 | 0.78 |
| | Baselines | $a_E$ | 0.90 | 0.77 | 0.86 | 0.44 | 0.80 | 0.83 | 0.80 | 1.00 |
| | | $a_L$ | 0.86 | 0.73 | 0.82 | 0.47 | 0.76 | 0.89 | 0.77 | 0.74 |
| | | $a_{MSP}$ | 0.90 | 0.80 | 0.87 | 0.41 | 0.81 | 0.77 | 0.80 | 1.00 |
| $s_1$ | Ours | $a_{D_\alpha^*}$ | 0.88 | **0.85** | **0.88** | **0.29** | 0.81 | **0.54** | 0.80 | 0.82 |
| | | $a_{D_{KL}^*}$ | 0.89 | 0.83 | **0.88** | 0.32 | 0.81 | 0.59 | 0.80 | 0.82 |
| | | $a_{D_\alpha^{diff}}$ | 0.79 | 0.70 | 0.76 | 0.45 | 0.67 | 0.85 | 0.75 | 0.60 |
| | | $a_{D_{KL}^{diff}}$ | 0.79 | 0.68 | 0.75 | 0.47 | 0.66 | 0.89 | 0.75 | 0.58 |
| | | $a_{D_\alpha^{mean}}$ | **0.90** | 0.77 | 0.86 | 0.44 | 0.79 | 0.83 | 0.80 | 0.79 |
| | | $a_{D_{KL}^{mean}}$ | 0.88 | 0.75 | 0.84 | 0.45 | 0.77 | 0.85 | 0.79 | 0.75 |
| | | $a_{FR^*}$ | 0.81 | 0.66 | 0.76 | 0.50 | 0.70 | 0.94 | 0.76 | 0.65 |
| | | $a_{FR^{diff}}$ | 0.52 | 0.51 | 0.52 | 0.50 | 0.33 | 0.95 | 0.54 | 0.24 |
| | | $a_{FR^{mean}}$ | 0.87 | 0.70 | 0.81 | 0.49 | 0.75 | 0.94 | 0.78 | 0.72 |
| | Baselines | $a_C$ | 0.67 | 0.59 | 0.64 | 0.49 | 0.00 | 0.94 | 0.00 | 0.00 |
| | | $a_M$ | 0.81 | 0.83 | 0.83 | 0.31 | 0.75 | 0.58 | 0.78 | 0.72 |

(d) es:news-EMEA

## C.3.2 DOMAIN SHIFTS

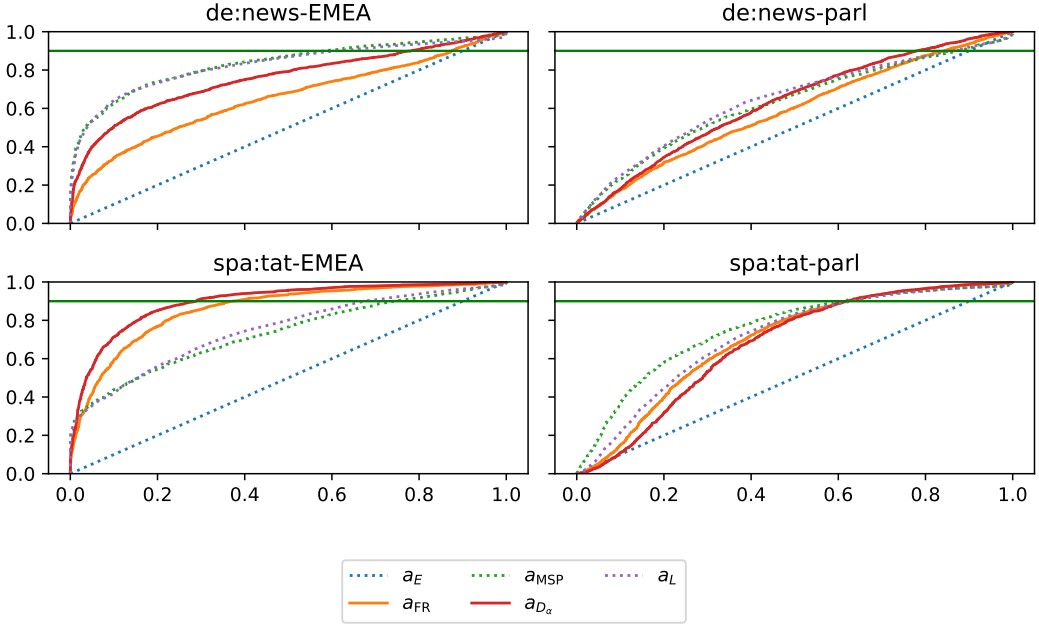

Figure 5: ROC-AUC curves for our uncertainty based metrics compared to common baselines for domain shifts detection. baselines are represented in dashed lines.

Table 10: Detailed performance results of our OOD detectors on dialog shift against the Multi WOZ dataset as reference set.

| Scenario | | Score | AUPR-IN | AUPR-OUT | AUROC | ERR | f1 | FPR | precision | recall |
|---|---|---|---|---|---|---|---|---|---|---|
| $s_0$ | Ours | $a_{D_\alpha}$ | **0.87** | **0.87** | **0.87** | **0.31** | 0.78 | **0.56** | 0.79 | 0.77 |
| | | $a_{FR}$ | 0.73 | 0.81 | 0.79 | 0.34 | 0.67 | 0.63 | 0.75 | 0.60 |
| | | $a_E$ | 0.75 | 0.75 | 0.62 | 0.40 | 0.67 | 0.75 | 0.53 | 1.00 |
| | Baselines | $a_L$ | 0.81 | 0.79 | 0.82 | 0.39 | 0.76 | 0.72 | 0.75 | 0.77 |
| | | $a_{MSP}$ | 0.53 | 0.64 | 0.57 | 0.42 | 0.32 | 0.78 | 0.53 | 0.22 |
| $s_1$ | | $a_{D^*}$ | 0.69 | 0.72 | 0.72 | 0.43 | 0.60 | 0.81 | 0.72 | 0.52 |
| | | $a_{D^*_{KL}}$ | 0.66 | 0.68 | 0.70 | 0.44 | 0.56 | 0.82 | 0.70 | 0.47 |
| | | $a_{D^{diff}}$ | 0.61 | 0.64 | 0.64 | 0.45 | 0.45 | 0.85 | 0.64 | 0.35 |
| | | $a_{D^{diff}_{KL}}$ | 0.62 | 0.63 | 0.65 | 0.46 | 0.47 | 0.87 | 0.65 | 0.37 |
| | Ours | $a_{D^{mean}}$ | 0.62 | 0.65 | 0.65 | 0.44 | 0.47 | 0.83 | 0.65 | 0.37 |
| | | $a_{D^{mean}_{KL}}$ | 0.64 | 0.65 | 0.67 | 0.46 | 0.51 | 0.86 | 0.67 | 0.41 |
| | | $a_{FR^*}$ | 0.69 | 0.69 | 0.72 | 0.43 | 0.60 | 0.82 | 0.71 | 0.51 |
| | | $a_{FR^{diff}}$ | 0.54 | 0.55 | 0.55 | 0.49 | 0.34 | 0.93 | 0.55 | 0.25 |
| | | $a_{FR^{mean}}$ | 0.55 | 0.57 | 0.57 | 0.48 | 0.37 | 0.91 | 0.58 | 0.27 |
| | Baselines | $a_C$ | 0.88 | 0.89 | 0.90 | 0.25 | 0.75 | 0.46 | 0.86 | 0.67 |
| | | $a_M$ | 0.87 | 0.84 | 0.87 | 0.33 | 0.80 | 0.61 | 0.80 | 0.80 |

(a) dailydialog-default

| Scenario | | Score | AUPR-IN | AUPR-OUT | AUROC | ERR | f1 | FPR | precision | recall |
|---|---|---|---|---|---|---|---|---|---|---|
| $s_0$ | Ours | $a_{D_\alpha}$ | 0.52 | **0.87** | **0.72** | **0.52** | 0.52 | **0.69** | 0.50 | 0.54 |
| | | $a_{FR}$ | 0.41 | 0.86 | 0.69 | 0.55 | 0.37 | 0.73 | 0.39 | 0.35 |
| | | $a_E$ | **0.63** | **0.87** | 0.63 | 0.56 | 0.43 | 0.74 | 0.28 | 1.00 |
| | Baselines | $a_L$ | 0.40 | 0.66 | 0.47 | 0.73 | 0.43 | 1.00 | 0.27 | 1.00 |
| | | $a_{MSP}$ | 0.36 | 0.86 | 0.67 | **0.52** | 0.31 | **0.69** | 0.34 | 0.28 |
| $s_1$ | | $a_{D^*}$ | 0.69 | 0.93 | 0.85 | 0.40 | 0.63 | 0.53 | 0.56 | 0.75 |
| | | $a_{D^*_{KL}}$ | 0.42 | 0.84 | 0.67 | 0.63 | 0.43 | 0.85 | 0.44 | 0.43 |
| | | $a_{D^{diff}}$ | 0.38 | 0.81 | 0.63 | 0.64 | 0.37 | 0.85 | 0.39 | 0.35 |
| | | $a_{D^{diff}_{KL}}$ | 0.36 | 0.80 | 0.62 | 0.66 | 0.36 | 0.89 | 0.39 | 0.34 |
| | Ours | $a_{D^{mean}}$ | 0.39 | 0.82 | 0.64 | 0.63 | 0.38 | 0.85 | 0.40 | 0.36 |
| | | $a_{D^{mean}_{KL}}$ | 0.39 | 0.81 | 0.64 | 0.67 | 0.40 | 0.90 | 0.41 | 0.38 |
| | | $a_{FR^*}$ | 0.53 | 0.88 | 0.75 | 0.58 | 0.54 | 0.77 | 0.51 | 0.59 |
| | | $a_{FR^{diff}}$ | 0.31 | 0.78 | 0.57 | 0.68 | 0.29 | 0.91 | 0.32 | 0.26 |
| | | $a_{FR^{mean}}$ | 0.31 | 0.80 | 0.58 | 0.66 | 0.29 | 0.88 | 0.33 | 0.26 |
| | Baselines | $a_C$ | 0.54 | 0.89 | 0.70 | 0.74 | 0.59 | 0.91 | 0.55 | 0.64 |
| | | $a_M$ | 0.53 | 0.87 | 0.74 | 0.73 | 0.56 | 0.99 | 0.53 | 0.60 |

(b) silicone-melds

| Scenario | | Score | AUPR-IN | AUPR-OUT | AUROC | ERR | f1 | FPR | precision | recall |
|---|---|---|---|---|---|---|---|---|---|---|
| $s_0$ | Ours | $a_{D_\alpha}$ | 0.52 | **0.87** | **0.72** | **0.52** | 0.52 | **0.69** | 0.50 | 0.54 |
| | | $a_{FR}$ | 0.41 | 0.86 | 0.69 | 0.55 | 0.37 | 0.73 | 0.39 | 0.35 |
| | | $a_E$ | **0.63** | **0.87** | 0.63 | 0.56 | 0.43 | 0.74 | 0.28 | 1.00 |
| | Baselines | $a_L$ | 0.40 | 0.66 | 0.47 | 0.73 | 0.43 | 1.00 | 0.27 | 1.00 |
| | | $a_{MSP}$ | 0.36 | 0.86 | 0.67 | **0.52** | 0.31 | **0.69** | 0.34 | 0.28 |
| $s_1$ | | $a_{D^*}$ | 0.69 | 0.93 | 0.85 | 0.40 | 0.63 | 0.53 | 0.56 | 0.75 |
| | | $a_{D^*_{KL}}$ | 0.42 | 0.84 | 0.67 | 0.63 | 0.43 | 0.85 | 0.44 | 0.43 |
| | | $a_{D^{diff}}$ | 0.38 | 0.82 | 0.64 | 0.64 | 0.38 | 0.86 | 0.40 | 0.36 |
| | | $a_{D^{diff}_{KL}}$ | 0.38 | 0.81 | 0.63 | 0.65 | 0.37 | 0.87 | 0.39 | 0.35 |
| | Ours | $a_{D^{mean}}$ | 0.39 | 0.83 | 0.65 | 0.63 | 0.39 | 0.84 | 0.41 | 0.38 |
| | | $a_{D^{mean}_{KL}}$ | 0.40 | 0.82 | 0.65 | 0.65 | 0.40 | 0.88 | 0.41 | 0.38 |
| | | $a_{FR^*}$ | 0.53 | 0.88 | 0.75 | 0.58 | 0.54 | 0.77 | 0.51 | 0.59 |
| | | $a_{FR^{diff}}$ | 0.31 | 0.77 | 0.57 | 0.69 | 0.30 | 0.93 | 0.34 | 0.28 |
| | | $a_{FR^{mean}}$ | 0.32 | 0.79 | 0.59 | 0.68 | 0.32 | 0.91 | 0.35 | 0.29 |
| | Baselines | $a_C$ | 0.54 | 0.89 | 0.70 | 0.74 | 0.59 | 0.91 | 0.55 | 0.64 |
| | | $a_M$ | 0.53 | 0.87 | 0.74 | 0.73 | 0.56 | 0.99 | 0.53 | 0.60 |

(c) silicone-melde

| Scenario | | Score | AUPR-IN | AUPR-OUT | AUROC | ERR | f1 | FPR | precision | recall |
|---|---|---|---|---|---|---|---|---|---|---|
| $s_0$ | Ours | $a_{D_\alpha}$ | 0.79 | **0.80** | 0.81 | **0.36** | 0.73 | **0.68** | 0.78 | 0.69 |
| | | $a_{FR}$ | 0.68 | 0.73 | 0.70 | 0.39 | 0.49 | 0.74 | 0.66 | 0.39 |
| | | $a_E$ | 0.75 | 0.75 | 0.64 | 0.45 | 0.67 | 0.84 | 0.61 | 1.00 |
| | Baselines | $a_L$ | **0.91** | 0.73 | **0.85** | 0.49 | 0.78 | 0.94 | 0.76 | 0.81 |
| | | $a_{MSP}$ | 0.70 | 0.65 | 0.64 | 0.43 | 0.54 | 0.80 | 0.69 | 0.45 |
| $s_1$ | | $a_{D^*}$ | 0.92 | 0.91 | 0.91 | 0.23 | 0.83 | 0.42 | 1.00 | 0.87 |
| | | $a_{D^*_{KL}}$ | 0.60 | 0.60 | 0.61 | 0.47 | 0.44 | 0.89 | 0.63 | 0.34 |
| | | $a_{D^{diff}}$ | 0.58 | 0.58 | 0.59 | 0.48 | 0.39 | 0.90 | 0.59 | 0.29 |
| | | $a_{D^{diff}_{KL}}$ | 0.58 | 0.57 | 0.59 | 0.48 | 0.41 | 0.91 | 0.61 | 0.31 |
| | Ours | $a_{D^{mean}}$ | 0.59 | 0.59 | 0.60 | 0.47 | 0.41 | 0.90 | 0.61 | 0.31 |
| | | $a_{D^{mean}_{KL}}$ | 0.59 | 0.58 | 0.59 | 0.48 | 0.41 | 0.90 | 0.61 | 0.31 |
| | | $a_{FR^*}$ | 0.75 | 0.74 | 0.75 | 0.41 | 0.65 | 0.77 | 0.74 | 0.58 |
| | | $a_{FR^{diff}}$ | 0.55 | 0.55 | 0.56 | 0.49 | 0.37 | 0.92 | 0.58 | 0.27 |
| | | $a_{FR^{mean}}$ | 0.56 | 0.59 | 0.59 | 0.47 | 0.39 | 0.89 | 0.59 | 0.29 |
| | Baselines | $a_C$ | 0.87 | 0.87 | 0.88 | 0.15 | 0.80 | 0.21 | 0.80 | 0.84 |
| | | $a_M$ | 0.85 | 0.93 | 0.91 | 0.10 | 0.83 | 0.21 | 0.81 | 0.85 |

(d) silicone-dydae

| Scenario | | Score | AUPR-IN | AUPR-OUT | AUROC | ERR | f1 | FPR | precision | recall |
|---|---|---|---|---|---|---|---|---|---|---|
| $s_0$ | Ours | $a_{D_\alpha}$ | 0.71 | 0.74 | **0.72** | **0.36** | 0.63 | 0.67 | 0.73 | 0.55 |
| | | $a_{FR}$ | 0.68 | 0.73 | **0.72** | **0.36** | 0.57 | 0.67 | 0.70 | 0.48 |
| | | $a_E$ | **0.75** | **0.75** | 0.66 | 0.39 | 0.67 | 0.74 | 0.53 | 1.00 |
| | Baselines | $a_L$ | 0.69 | 0.48 | 0.57 | 0.50 | 0.67 | 1.00 | 0.50 | 1.00 |
| | | $a_{MSP}$ | 0.61 | 0.74 | 0.71 | **0.36** | 0.00 | **0.66** | 0.00 | 0.00 |
| $s_1$ | | $a_{D^*}$ | 0.88 | 0.89 | 0.88 | 0.24 | 0.80 | 0.44 | 0.79 | 0.82 |
| | | $a_{D^*_{KL}}$ | 0.68 | 0.70 | 0.70 | 0.42 | 0.56 | 0.80 | 0.70 | 0.47 |
| | | $a_{D^{diff}}$ | 0.63 | 0.65 | 0.65 | 0.45 | 0.48 | 0.85 | 0.66 | 0.38 |
| | | $a_{D^{diff}_{KL}}$ | 0.63 | 0.64 | 0.65 | 0.45 | 0.48 | 0.85 | 0.65 | 0.38 |
| | Ours | $a_{D^{mean}}$ | 0.64 | 0.67 | 0.67 | 0.43 | 0.49 | 0.82 | 0.66 | 0.39 |
| | | $a_{D^{mean}_{KL}}$ | 0.65 | 0.66 | 0.67 | 0.44 | 0.51 | 0.84 | 0.67 | 0.41 |
| | | $a_{FR^*}$ | 0.79 | 0.80 | 0.80 | 0.36 | 0.71 | 0.68 | 0.76 | 0.66 |
| | | $a_{FR^{diff}}$ | 0.55 | 0.56 | 0.56 | 0.48 | 0.36 | 0.92 | 0.57 | 0.26 |
| | | $a_{FR^{mean}}$ | 0.57 | 0.59 | 0.59 | 0.47 | 0.38 | 0.89 | 0.58 | 0.28 |
| | Baselines | $a_C$ | 0.87 | 0.87 | 0.88 | 0.15 | 0.80 | 0.21 | 0.80 | 0.84 |
| | | $a_M$ | 0.87 | 0.95 | 0.93 | 0.10 | 0.86 | 0.21 | 0.82 | 0.90 |

(e) silicone-swda

| Scenario | | Score | AUPR-IN | AUPR-OUT | AUROC | ERR | f1 | FPR | precision | recall |
|---|---|---|---|---|---|---|---|---|---|---|
| $s_0$ | Ours | $a_{D_\alpha}$ | 0.79 | **0.80** | 0.81 | **0.36** | 0.73 | **0.68** | 0.78 | 0.69 |
| | | $a_{FR}$ | 0.68 | 0.73 | 0.70 | 0.39 | 0.49 | 0.74 | 0.66 | 0.39 |
| | | $a_E$ | 0.75 | 0.75 | 0.64 | 0.45 | 0.67 | 0.84 | 0.61 | 1.00 |
| | Baselines | $a_L$ | **0.91** | 0.73 | **0.85** | 0.49 | 0.78 | 0.94 | 0.76 | 0.81 |
| | | $a_{MSP}$ | 0.70 | 0.65 | 0.64 | 0.43 | 0.54 | 0.80 | 0.69 | 0.45 |
| $s_1$ | | $a_{D^*}$ | 0.92 | 0.91 | 0.91 | 0.23 | 0.83 | 0.42 | 1.00 | 0.87 |
| | | $a_{D^*_{KL}}$ | 0.60 | 0.60 | 0.61 | 0.47 | 0.44 | 0.89 | 0.63 | 0.34 |
| | | $a_{D^{diff}}$ | 0.58 | 0.58 | 0.59 | 0.48 | 0.40 | 0.90 | 0.60 | 0.30 |
| | | $a_{D^{diff}_{KL}}$ | 0.57 | 0.56 | 0.58 | 0.48 | 0.40 | 0.92 | 0.60 | 0.30 |
| | Ours | $a_{D^{mean}}$ | 0.59 | 0.59 | 0.60 | 0.47 | 0.41 | 0.90 | 0.61 | 0.31 |
| | | $a_{D^{mean}_{KL}}$ | 0.58 | 0.57 | 0.59 | 0.49 | 0.41 | 0.92 | 0.61 | 0.31 |
| | | $a_{FR^*}$ | 0.75 | 0.74 | 0.75 | 0.41 | 0.65 | 0.77 | 0.74 | 0.58 |
| | | $a_{FR^{diff}}$ | 0.54 | 0.56 | 0.56 | 0.49 | 0.35 | 0.92 | 0.56 | 0.28 |
| | | $a_{FR^{mean}}$ | 0.55 | 0.58 | 0.58 | 0.47 | 0.38 | 0.89 | 0.58 | 0.28 |
| | Baselines | $a_C$ | 0.84 | 0.87 | 0.87 | 0.13 | 0.81 | 0.22 | 0.81 | 0.84 |
| | | $a_M$ | 0.85 | 0.93 | 0.91 | 0.10 | 0.83 | 0.21 | 0.81 | 0.85 |

(f) silicone-dydada

| Scenario | | Score | AUPR-IN | AUPR-OUT | AUROC | ERR | f1 | FPR | precision | recall |
|---|---|---|---|---|---|---|---|---|---|---|
| $s_0$ | Ours | $a_{D_\alpha}$ | 0.42 | **0.91** | **0.72** | 0.57 | 0.45 | **0.70** | 0.40 | 0.50 |
| | | $a_{FR}$ | 0.41 | 0.90 | **0.72** | 0.57 | 0.38 | 0.71 | 0.35 | 0.41 |
| | | $a_E$ | **0.61** | 0.89 | 0.63 | 0.61 | 0.35 | 0.76 | 0.23 | 1.00 |
| | Baselines | $a_L$ | 0.53 | 0.76 | 0.58 | 0.80 | 0.40 | 1.00 | 0.34 | 0.48 |
| | | $a_{MSP}$ | 0.35 | 0.90 | 0.64 | **0.56** | 0.33 | **0.70** | 0.32 | 0.35 |
| $s_1$ | | $a_{D^*}$ | 0.67 | 0.95 | 0.86 | 0.44 | 0.58 | 0.55 | 0.48 | 0.77 |
| | | $a_{D^*_{KL}}$ | 0.29 | 0.85 | 0.63 | 0.72 | 0.34 | 0.90 | 0.32 | 0.36 |
| | | $a_{D^{diff}}$ | 0.27 | 0.84 | 0.60 | 0.72 | 0.30 | 0.90 | 0.29 | 0.31 |
| | | $a_{D^{diff}_{KL}}$ | 0.27 | 0.84 | 0.59 | 0.70 | 0.29 | 0.88 | 0.28 | 0.29 |
| | Ours | $a_{D^{mean}}$ | 0.27 | 0.85 | 0.61 | 0.72 | 0.30 | 0.89 | 0.29 | 0.31 |
| | | $a_{D^{mean}_{KL}}$ | 0.28 | 0.85 | 0.61 | 0.71 | 0.30 | 0.89 | 0.29 | 0.31 |
| | | $a_{FR^*}$ | 0.41 | 0.90 | 0.73 | 0.61 | 0.46 | 0.77 | 0.41 | 0.52 |
| | | $a_{FR^{diff}}$ | 0.23 | 0.82 | 0.54 | 0.74 | 0.23 | 0.92 | 0.23 | 0.22 |
| | | $a_{FR^{mean}}$ | 0.24 | 0.83 | 0.56 | 0.73 | 0.24 | 0.91 | 0.25 | 0.24 |
| | Baselines | $a_C$ | 0.45 | 0.92 | 0.75 | 0.78 | 0.53 | 0.97 | 0.50 | 0.56 |
| | | $a_M$ | 0.42 | 0.89 | 0.70 | 0.78 | 0.47 | 0.99 | 0.42 | 0.54 |

(g) silicone-iemocap

| Scenario | | Score | AUPR-IN | AUPR-OUT | AUROC | ERR | f1 | FPR | precision | recall |
|---|---|---|---|---|---|---|---|---|---|---|
| $s_0$ | Ours | $a_{D_\alpha}$ | 0.71 | **0.77** | **0.75** | 0.33 | 0.60 | 0.60 | 0.72 | 0.52 |
| | | $a_{FR}$ | 0.69 | 0.74 | 0.74 | 0.36 | 0.58 | 0.67 | 0.71 | 0.49 |
| | | $a_E$ | **0.75** | 0.75 | 0.70 | 0.37 | 0.67 | 0.68 | 0.60 | 1.00 |
| | Baselines | $a_L$ | 0.72 | 0.76 | 0.58 | 0.57 | 0.50 | 0.67 | 1.00 | 0.50 |
| | | $a_{MSP}$ | 0.62 | 0.76 | 0.73 | **0.32** | 0.00 | **0.59** | 0.00 | 0.00 |
| $s_1$ | | $a_{D^*}$ | 0.86 | 0.88 | 0.87 | 0.23 | 0.79 | 0.42 | 0.79 | 0.80 |
| | | $a_{D^*_{KL}}$ | 0.66 | 0.71 | 0.70 | 0.41 | 0.52 | 0.77 | 0.68 | 0.42 |
| | | $a_{D^{diff}}$ | 0.61 | 0.66 | 0.64 | 0.43 | 0.45 | 0.82 | 0.63 | 0.34 |
| | | $a_{D^{diff}_{KL}}$ | 0.62 | 0.64 | 0.64 | 0.44 | 0.46 | 0.84 | 0.64 | 0.36 |
| | Ours | $a_{D^{mean}}$ | 0.63 | 0.68 | 0.66 | 0.42 | 0.45 | 0.79 | 0.64 | 0.35 |
| | | $a_{D^{mean}_{KL}}$ | 0.63 | 0.65 | 0.66 | 0.44 | 0.48 | 0.84 | 0.66 | 0.38 |
| | | $a_{FR^*}$ | 0.77 | 0.80 | 0.79 | 0.36 | 0.69 | 0.66 | 0.75 | 0.63 |
| | | $a_{FR^{diff}}$ | 0.55 | 0.57 | 0.57 | 0.48 | 0.36 | 0.91 | 0.57 | 0.27 |
| | | $a_{FR^{mean}}$ | 0.57 | 0.59 | 0.59 | 0.47 | 0.40 | 0.90 | 0.60 | 0.30 |
| | Baselines | $a_C$ | 0.87 | 0.92 | 0.94 | 0.13 | 0.89 | 0.22 | 0.82 | 0.65 |
| | | $a_M$ | 0.88 | 0.96 | 0.94 | 0.10 | 0.88 | 0.21 | 0.82 | 0.94 |

(h) silicone-mrda

de:news-EMEA

de:news-parl

spa:tat-EMEA

spa:tat-parl

23

Legend: $a_{FR^*}$, $a_{D_{KL}^*}$, $a_{D_\alpha^*}$, $a_{D_\alpha^*}$, $a_{D_\alpha^*}$, $a_{D_\alpha^*}$, $a_M$

### C.3.3 DIALOG SHIFTS

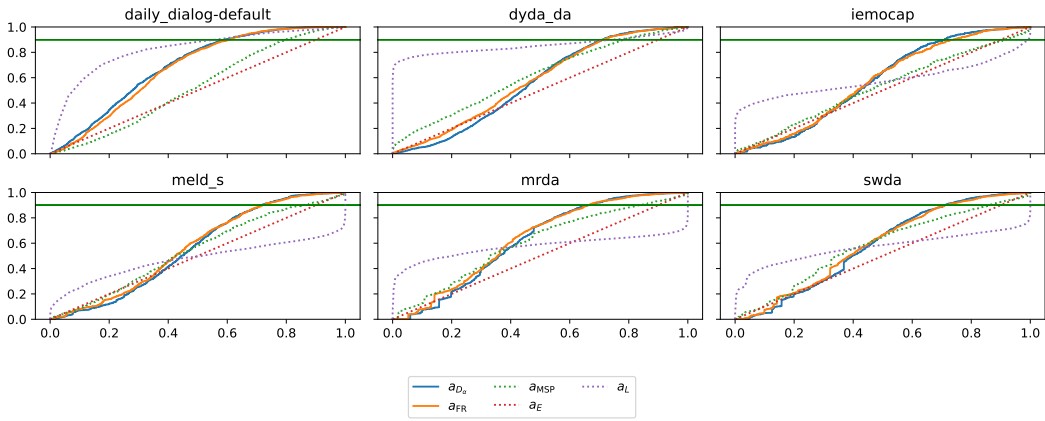

Figure 7: ROC-AUC curves for our uncertainty based metrics compared to common baselines for dialog shifts detection. baselines are represented in dashed lines.

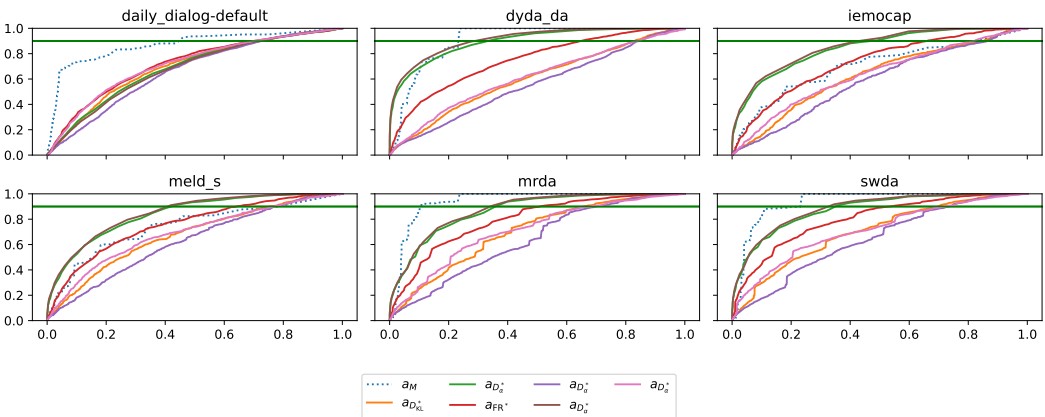

Figure 8: Rocauc curves for our reference based metrics compared to common baselines for dialog shift detection. baselines are represented in dashed lines.

## D NTM PERFORMANCE

Surprisingly we show that common OOD detectors tend to exclude samples that are well handled by the model and keep some that are not leading to decreasing overall performance in terms of translation metrics. Moreoever it seems this phenomenon is more dominant in reference based detectors. We show that our uncertainty based detectors mostly avoir that downfall and provide good OOD detection and improved translation performances.

### D.1 ABSOLUTE PERFORMANCES

It is clear (somewhat expected) that NMT models do not perform as well on OOD data as we can see in Tab. 11b. However, we find that our OOD detectors are able to remove most of the worst case samples and keep enough well translated samples so that with correct filtering our method actually allow the model to achieve somewhat acceptable BLEU scores.

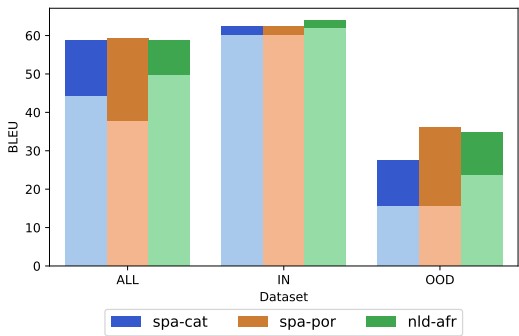

Figure 9: Gain in translation performances when filtering OOD samples with our method on different datasets and language pairs.

| Scenario | | Score | spa-cat | spa-por | nld-afr | spa:tat-parl | de:news-parl | spa:tat-EMEA | de:news-EMEA |
|---|---|---|---|---|---|---|---|---|---|
| | | $\chi$ | 59.73 | 59.73 | 61.19 | 59.73 | 34.07 | 59.73 | 34.07 |
| $\mathsf{S}_0$ | Ours | $a_{D_\alpha}$ | 62.48 | 62.48 | 66.43 | 64.92 | 36.68 | 63.86 | 36.69 |
| | | $a_{FR}$ | 62.84 | 62.84 | 64.89 | 63.75 | 36.10 | 62.84 | 36.19 |
| | Baselines | $a_E$ | 62.66 | 62.66 | 64.53 | 62.66 | 35.87 | 62.66 | 35.87 |
| | | $a_L$ | 64.77 | 64.77 | 66.60 | 64.77 | 36.72 | 64.77 | 36.72 |
| | | $a_{MSP}$ | 58.63 | 58.63 | 59.59 | 58.63 | 33.00 | 58.63 | 33.00 |
| $\mathsf{S}_1$ | Ours | $a_{D_\alpha^*}$ | 60.43 | 60.43 | 61.74 | 60.43 | 33.82 | 60.43 | 33.82 |
| | | $a_{FR^*}$ | 60.47 | 60.47 | 61.73 | 60.47 | 33.78 | 60.47 | 33.78 |
| | Baselines | $a_C$ | 60.78 | 60.78 | 61.04 | 60.78 | 34.48 | 60.78 | 34.48 |
| | | $a_M$ | 59.87 | 59.87 | 61.36 | 59.87 | 33.85 | 59.87 | 33.85 |

(a) IN

| Scenario | | Score | spa-cat | spa-por | nld-afr | spa:tat-parl | de:news-parl | spa:tat-EMEA | de:news-EMEA |
|---|---|---|---|---|---|---|---|---|---|
| | | $\chi$ | 15.73 | 15.40 | 23.79 | 33.17 | 28.36 | 59.38 | 52.16 |
| $\mathsf{S}_0$ | Ours | $a_{D_\alpha}$ | 27.22 | 36.88 | 35.80 | 41.70 | 33.64 | 64.11 | 54.60 |
| | | $a_{FR}$ | 26.26 | 36.99 | 33.00 | 38.17 | 32.10 | 65.24 | 55.02 |
| | Baselines | $a_E$ | 14.43 | 12.98 | 27.71 | 35.07 | 31.97 | 60.50 | 54.48 |
| | | $a_L$ | 16.62 | 14.70 | 33.59 | 40.23 | 34.18 | 61.32 | 54.41 |
| | | $a_{MSP}$ | 17.36 | 18.97 | 23.91 | 30.81 | 26.46 | 41.51 | 34.64 |
| $\mathsf{S}_1$ | Ours | $a_{D_\alpha^*}$ | 19.66 | 22.81 | 33.72 | 34.05 | 27.38 | 46.25 | 43.62 |
| | | $a_{FR^*}$ | 19.72 | 22.89 | 33.82 | 34.12 | 27.39 | 45.82 | 43.61 |
| | Baselines | $a_C$ | 16.84 | 16.72 | 23.67 | 33.32 | 29.11 | 59.55 | 52.19 |
| | | $a_M$ | 16.55 | 19.30 | 25.15 | 30.75 | 27.22 | 59.15 | 54.71 |

(b) OOD

| Scenario | | Score | spa-cat | spa-por | nld-afr | spa:tat-parl | de:news-parl | spa:tat-EMEA | de:news-EMEA |
|---|---|---|---|---|---|---|---|---|---|
| | | $\chi$ | 44.24 | 37.57 | 49.45 | 46.45 | 31.21 | 59.55 | 43.12 |
| $\mathsf{S}_0$ | Ours | $a_{D_\alpha}$ | 59.52 | 60.15 | 62.53 | 56.42 | 35.35 | 63.98 | 46.32 |
| | | $a_{FR}$ | 59.50 | 60.50 | 59.87 | 52.82 | 34.34 | 63.39 | 46.11 |
| | Baselines | $a_E$ | 48.75 | 40.58 | 57.89 | 48.77 | 34.19 | 61.48 | 45.78 |
| | | $a_L$ | 54.37 | 48.54 | 61.70 | 54.76 | 35.64 | 62.92 | 46.22 |
| | | $a_{MSP}$ | 46.62 | 42.62 | 47.75 | 44.21 | 29.50 | 56.16 | 33.37 |
| $\mathsf{S}_1$ | Ours | $a_{D_\alpha^*}$ | 49.72 | 48.34 | 57.76 | 56.97 | 30.37 | 59.52 | 36.91 |
| | | $a_{FR^*}$ | 49.81 | 48.40 | 57.72 | 57.03 | 30.37 | 59.53 | 36.89 |
| | Baselines | $a_C$ | 47.98 | 42.07 | 50.03 | 45.97 | 31.77 | 60.12 | 43.84 |
| | | $a_M$ | 48.70 | 49.44 | 52.91 | 49.61 | 30.42 | 59.84 | 37.86 |

(c) ALL

Table 11: Absolue translation performances in terms of BLEU on the different subset (IN, OOD, ALL) of each dataset of our translation OOD performance benchmark.

| Scenario | | Score | spa-cat | spa-por | nld-afr | spa:tat-parl | de:news-parl | spa:tat-EMEA | de:news-EMEA |
|---|---|---|---|---|---|---|---|---|---|
| $s_0$ | Ours | $a_{D_\alpha}$ | +2.75 | +2.75 | +5.24 | +5.19 | +2.61 | +4.13 | +2.62 |
| | | $a_{FR}$ | +3.11 | +3.11 | +3.70 | +4.02 | +2.03 | +3.11 | +2.11 |
| | Baselines | $a_E$ | +2.93 | +2.93 | +3.34 | +2.93 | +1.79 | +2.93 | +1.79 |
| | | $a_L$ | +5.03 | +5.03 | +5.41 | +5.03 | +2.65 | +5.03 | +2.65 |
| | | $a_{MSP}$ | -1.10 | -1.10 | -1.60 | -1.10 | -1.07 | -1.10 | -1.07 |
| $s_1$ | Ours | $a_{D_\alpha^*}$ | +0.70 | +0.70 | +0.55 | +0.70 | -0.26 | +0.70 | -0.26 |
| | | $a_{FR^*}$ | +0.74 | +0.74 | +0.54 | +0.74 | -0.29 | +0.74 | -0.29 |
| | Baselines | $a_C$ | +1.05 | +1.05 | -0.15 | +1.05 | +0.40 | +1.05 | +0.40 |
| | | $a_M$ | +0.14 | +0.14 | +0.17 | +0.14 | -0.22 | +0.14 | -0.22 |

(a) IN

| Scenario | | Score | spa-cat | spa-por | nld-afr | spa:tat-parl | de:news-parl | spa:tat-EMEA | de:news-EMEA |
|---|---|---|---|---|---|---|---|---|---|
| $s_0$ | Ours | $a_{D_\alpha}$ | +11.49 | +21.48 | +12.02 | +8.52 | +5.28 | +4.74 | +2.44 |
| | | $a_{FR}$ | +10.53 | +21.59 | +9.22 | +5.00 | +3.75 | +5.87 | +2.85 |
| | Baselines | $a_E$ | -1.30 | -2.42 | +3.92 | +1.89 | +3.61 | +1.12 | +2.31 |
| | | $a_L$ | +0.89 | -0.70 | +9.80 | +7.06 | +5.82 | +1.94 | +2.24 |
| | | $a_{MSP}$ | +1.63 | +3.57 | +0.12 | -2.36 | -1.90 | -17.87 | -17.52 |
| $s_1$ | Ours | $a_{D_\alpha^*}$ | +3.94 | +7.41 | +9.93 | +0.88 | -0.98 | -13.13 | -8.54 |
| | | $a_{FR^*}$ | +3.99 | +7.49 | +10.03 | +0.95 | -0.97 | -13.56 | -8.56 |
| | Baselines | $a_C$ | +1.11 | +1.32 | -0.12 | +0.14 | +0.75 | +0.17 | +0.02 |
| | | $a_M$ | +0.82 | +3.90 | +1.36 | -2.43 | -1.14 | -0.22 | +2.55 |

(b) OOD

| Scenario | | Score | spa-cat | spa-por | nld-afr | spa:tat-parl | de:news-parl | spa:tat-EMEA | de:news-EMEA |
|---|---|---|---|---|---|---|---|---|---|
| $s_0$ | Ours | $a_{D_\alpha}$ | +15.28 | +22.58 | +13.09 | +9.97 | +4.14 | +4.43 | +3.20 |
| | | $a_{FR}$ | +15.26 | +22.93 | +10.42 | +6.37 | +3.13 | +3.84 | +2.99 |
| | Baselines | $a_E$ | +4.51 | +3.01 | +8.44 | +2.32 | +2.98 | +1.92 | +2.66 |
| | | $a_L$ | +10.13 | +10.98 | +12.26 | +8.31 | +4.42 | +3.36 | +3.10 |
| | | $a_{MSP}$ | +2.38 | +5.06 | -1.70 | -2.24 | -1.72 | -3.40 | -9.74 |
| $s_1$ | Ours | $a_{D_\alpha^*}$ | +5.48 | +10.78 | +8.31 | +10.52 | -0.85 | -0.04 | -6.20 |
| | | $a_{FR^*}$ | +5.57 | +10.83 | +8.27 | +10.58 | -0.85 | -0.02 | -6.23 |
| | Baselines | $a_C$ | +3.74 | +4.51 | +0.58 | -0.49 | +0.56 | +0.57 | +0.72 |
| | | $a_M$ | +4.46 | +11.87 | +3.46 | +3.16 | -0.79 | +0.28 | -5.26 |

(c) ALL

| Scenario | | Dataset Score | spa-cat | | | spa-por | | | nld-afr | | | spa:tat-parl | | | de:news-parl | | | spa:tat-EMEA | | | de:news-EMEA | | |
|---|---|---|---|---|---|---|---|---|---|---|---|---|---|---|---|---|---|---|---|---|---|---|---|
| | | | ALL | IN | OOD | ALL | IN | OOD | ALL | IN | OOD | ALL | IN | OOD | ALL | IN | OOD | ALL | IN | OOD | ALL | IN | OOD |
| $s_0$ | Ours | $a_{D_\alpha}$ | 43% | 20% | 86% | 56% | 20% | 92% | 37% | 20% | 75% | 37% | 20% | 54% | 29% | 20% | 38% | 20% | 19% | 20% | 14% | 20% | 7% |
| | | $a_{FR}$ | 43% | 20% | 85% | 56% | 20% | 92% | 32% | 16% | 66% | 28% | 18% | 39% | 28% | 20% | 37% | 48% | 20% | 76% | 15% | 19% | 10% |
| | Baselines | $a_E$ | 27% | 20% | 40% | 28% | 20% | 36% | 33% | 20% | 62% | 19% | 20% | 19% | 30% | 20% | 40% | 12% | 20% | 4% | 14% | 20% | 9% |
| | | $a_L$ | 33% | 19% | 59% | 40% | 19% | 61% | 35% | 19% | 69% | 32% | 19% | 44% | 30% | 19% | 40% | 13% | 19% | 6% | 13% | 19% | 6% |
| | | $a_{MSP}$ | 26% | 19% | 38% | 32% | 19% | 45% | 14% | 16% | 9% | 15% | 19% | 12% | 14% | 20% | 8% | 52% | 19% | 86% | 48% | 20% | 76% |
| $s_1$ | Ours | $a_{D_\alpha^*}$ | 29% | 19% | 47% | 41% | 19% | 62% | 36% | 20% | 71% | 54% | 19% | 88% | 12% | 19% | 6% | 57% | 19% | 94% | 40% | 20% | 62% |
| | | $a_{FR^*}$ | 30% | 20% | 48% | 41% | 20% | 62% | 36% | 20% | 71% | 54% | 20% | 88% | 12% | 18% | 6% | 57% | 20% | 94% | 40% | 18% | 62% |
| | Baselines | $a_C$ | 23% | 16% | 37% | 27% | 16% | 38% | 17% | 15% | 22% | 9% | 16% | 2% | 10% | 11% | 10% | 9% | 16% | 2% | 6% | 11% | 0% |
| | | $a_M$ | 30% | 20% | 49% | 46% | 20% | 72% | 28% | 20% | 47% | 38% | 20% | 56% | 17% | 20% | 14% | 58% | 20% | 96% | 50% | 20% | 81% |

Table 13: Share of the datasets removed when taking $\gamma$ so that we keep 80% of the IN distribution.

## D.2 GAINS

## D.3 EFFECT OF A LARGER THRESHOLD ON NMT PERFORMANCE

Table 15: Detailed impacts on NMT performance results per tasks (Domain- or Language-shifts) of the different OOD detectors with a threshold defined to keep 99% of the IN data. We present results on the different part of the data: IN data, OOD data and the combination of both, ALL. For each we report the absolute average BLEU score (Abs.), the average gains in BLEU (G.s.) compared to a setting without OOD filtering ($f_\theta$ only) and the share of the subset removed by the detector (R.Sh.).

| | | Domain shifts | | | | | | | | | Language shifts | | | | | | | | |
| | | IN | | | OOD | | | ALL | | | IN | | | OOD | | | ALL | | |
| | | Abs. | G.s | R.Sh | Abs. | G.s | R.Sh | Abs. | G.s | R.Sh | Abs. | G.s | R.Sh | Abs. | G.s | R.Sh | Abs. | G.s | R.Sh |
|---|---|---|---|---|---|---|---|---|---|---|---|---|---|---|---|---|---|---|---|
| | ✗ | 47.1 | +0.0 | 0.0% | 43.4 | +0.0 | 0.0% | 45.3 | +0.0 | 0.0% | 60.5 | +0.0 | 0.0% | 18.1 | +0.0 | 0.0% | 43.9 | +0.0 | 0.0% |
| $s_0$ Ours | $a_{D_\alpha}$ | 47.3 | +0.2 | 1.0% | 44.2 | +0.8 | 5.5% | 45.8 | +0.5 | 3.2% | 60.7 | +0.2 | 1.0% | 21.8 | +3.7 | 34.5% | 49.2 | +5.4 | 14.6% |
| | $a_{FR}$ | 47.3 | +0.2 | 1.0% | 44.1 | +0.7 | 7.2% | 45.7 | +0.4 | 4.1% | 60.7 | +0.2 | 1.0% | 22.3 | +4.2 | 37.0% | 49.7 | +5.9 | 15.8% |
| Bas. | $a_E$ | 47.3 | +0.2 | 1.0% | 44.0 | +0.5 | 1.9% | 45.6 | +0.4 | 1.4% | 60.9 | +0.4 | 1.0% | 18.7 | +0.6 | 17.6% | 46.0 | +2.1 | 7.3% |
| | $a_L$ | 47.3 | +0.2 | 0.9% | 44.0 | +0.6 | 1.9% | 45.6 | +0.4 | 1.4% | 60.8 | +0.3 | 0.9% | 19.1 | +0.9 | 18.4% | 46.2 | +2.3 | 7.6% |
| | $a_{MSP}$ | 47.0 | -0.1 | 1.0% | 40.3 | -3.1 | 14.7% | 43.5 | -1.8 | 7.8% | 60.4 | -0.1 | 1.0% | 18.5 | +0.3 | 4.3% | 44.3 | +0.5 | 2.5% |
| $s_1$ Ours | $a_{D^*_\alpha}$ | 47.0 | -0.1 | 0.9% | 40.3 | -3.1 | 26.5% | 43.9 | -1.4 | 13.7% | 60.5 | -0.0 | 0.9% | 19.3 | +1.1 | 10.5% | 45.3 | +1.4 | 4.8% |
| | $a_{FR^*}$ | 47.0 | -0.1 | 0.9% | 40.3 | -3.1 | 26.6% | 43.9 | -1.3 | 13.8% | 60.5 | -0.0 | 0.9% | 19.3 | +1.1 | 10.5% | 45.3 | +1.4 | 4.8% |
| Bas. | $a_M$ | 47.0 | -0.1 | 1.0% | 41.6 | -1.8 | 18.1% | 44.6 | -0.7 | 9.6% | 60.5 | -0.0 | 1.0% | 18.4 | +0.3 | 12.1% | 45.3 | +1.4 | 5.9% |

# E NEGATIVE RESULTS

## E.1 DIFFERENT AGGREGATION OF OOD METRICS

Most of our detectors are initially classification OOD detectors that we adapted for text generation by averaging them over the generated sequences and using this aggregated score as a score for the whole sequence. We experimented with other aggregations such as the standard deviation or the min/max along the sequence. If the standard deviation gave relatively good results they were still less interesting that the naive average.

Table 14: Correlation between OOD scores and translation metrics BLEU and BERT−S

| | | Bertscore f1 | | | Bleu score | | |
| | | ALL | IN | OUT | ALL | IN | OUT |
| Scenario | Score | | | | | | |
|---|---|---|---|---|---|---|---|
| $s_0$ Ours | $a_{D_\alpha}$ | -0.49 | -0.33 | -0.53 | -0.37 | -0.27 | -0.38 |
| | $a_{FR}$ | -0.47 | -0.33 | -0.45 | -0.38 | -0.28 | -0.37 |
| Baselines | $a_E$ | -0.19 | -0.30 | 0.00 | -0.16 | -0.21 | 0.00 |
| | $a_L$ | -0.22 | -0.26 | 0.23 | -0.30 | -0.29 | -0.06 |
| | $a_{MSP}$ | -0.28 | -0.15 | -0.43 | -0.14 | -0.09 | -0.14 |
| $s_1$ Ours | $a_{D^*_\alpha}$ | -0.35 | -0.22 | -0.65 | -0.24 | -0.18 | -0.35 |
| | $a_{FR^*}$ | -0.34 | -0.22 | -0.64 | -0.24 | -0.18 | -0.37 |
| Baselines | $a_C$ | -0.12 | -0.07 | -0.05 | -0.11 | -0.06 | -0.12 |
| | $a_M$ | -0.23 | -0.14 | -0.29 | -0.11 | -0.06 | -0.05 |

## E.2 NEGENTROPY
### OF BAG OF DISTRIBUTIONS

We introduced in Sec. 3.3 the bag of distributions as a way to aggregate a sequence of probability distribution and compare it to a set of reference using information projections Sec. 3.3. A natural idea would be to apply the Negentropy methods (Sec. 3.2) to these aggregated distributions.

More formally given a sequence of probability distribution $\mathcal{S}_\theta(\mathbf{x}) = \{p_\theta^T(\mathbf{x}, \hat{\mathbf{y}}_{\leqslant t})\}_{t=1}^n$ we would compute its bag of distributions:

$$\bar{p}_\theta(x) \triangleq \frac{1}{|y|} \sum_{t=1}^{|y|} p_\theta(x, y_{\leqslant t}) \tag{7}$$

And then compute as novelty score:

$$J_D(\mathbf{p}) = D(\mathbf{p} \| \mathcal{U}) \tag{8}$$

Further experiments have shown that this process was unable to discriminate OOD samples or improve performance translation. We suspect that the uncertainty at each step is key to capture the behavior of the language model and that this uncertainty information is lost when averaging probability distribution along the sequence.

### E.3 DIFFERENT REFERENCE DISTRIBUTIONS

In the no-reference scenario we used the uniform distribution as reference distribution to compare against. However, we can obviously use other reference distributions. We tried two natural options: the tf-idf Yuan et al. (2021) distribution and the average distribution on the reference set. The latter effectively replacing the projection onto the reference set by the distance to the average element of it.

It is worth signaling that these methods falls into the reference scenario since we need it to compute these statistics. They would be interesting though if they could maintain performance while being less computationally expensive than the projection.

We found out that these references were not as efficient as the projection onto the reference set and did not achieve better performance than their no reference counterparts.

| | | | Language shifts | | | Domain shifts | | | Dialog shifts | | |
|---|---|---|---|---|---|---|---|---|---|---|---|
| | | | AUROC | FPR | F1 | AUROC | FPR | F1 | AUROC | FPR | F1 |
| $s_0$ | Ours | $a_{D_\alpha}$ | **0.95** | **0.25** | **0.84** | **0.85** | **0.62** | **0.75** | **0.79** | **0.64** | **0.66** |
| | | $a_{FR}$ | 0.93 | 0.28 | 0.83 | 0.74 | 0.87 | 0.60 | 0.72 | 0.70 | 0.64 |
| | Bas. | $a_E$ | 0.89 | 0.44 | 0.77 | 0.76 | 0.78 | 0.71 | 0.65 | 0.76 | 0.57 |
| | | $a_{MSP}$ | 0.87 | 0.44 | 0.75 | 0.78 | 0.77 | 0.71 | 0.66 | 0.72 | 0.21 |
| | | $a_L$ | 0.78 | 0.79 | 0.50 | 0.72 | 0.89 | 0.65 | 0.65 | 0.95 | 0.62 |
| $s_1$ | Ours | $a_{D^*}$ | 0.88 | 0.34 | 0.71 | **0.86** | **0.50** | **0.70** | **0.86** | **0.52** | **0.59** |
| | | $a_{FR^*}$ | 0.88 | 0.35 | 0.69 | 0.81 | 0.69 | 0.69 | 0.76 | 0.75 | 0.38 |
| | | $a_{D_\alpha^{mean}}$ | 0.84 | 0.41 | 0.62 | 0.68 | 0.84 | 0.67 | 0.64 | 0.86 | 0.57 |
| | | $a_{D_\alpha^{idf}}$ | 0.82 | 0.48 | 0.62 | 0.66 | 0.87 | 0.67 | 0.63 | 0.87 | 0.57 |
| | | $a_{FR^{mean}}$ | 0.72 | 0.72 | 0.62 | 0.53 | 0.94 | 0.66 | 0.58 | 0.90 | 0.57 |
| | | $a_{FR^{idf}}$ | 0.67 | 0.81 | 0.61 | 0.52 | 0.94 | 0.66 | 0.56 | 0.92 | 0.57 |
| | Bas. | $a_M$ | **0.92** | **0.26** | **0.73** | 0.78 | 0.59 | 0.40 | 0.84 | 0.55 | 0.56 |
| | | $a_C$ | 0.71 | 0.80 | 0.62 | 0.68 | 0.76 | 0.67 | 0.72 | 0.61 | 0.48 |

Table 16: Summary of the results including different custom reference distributions.

### E.4 IMPACT OF ADDITIONAL FINETUNING ON IN DATA

| | | | Language shifts | | | Domain shifts | | | Dialog shifts | | |
|---|---|---|---|---|---|---|---|---|---|---|---|
| | | | AUROC | FPR | F1 | AUROC | FPR | F1 | AUROC | FPR | F1 |
| $s_0$ | Ours | $a_{D_\alpha}$ | **0.94** | **0.26** | **0.80** | **0.86** | **0.65** | **0.70** | **0.79** | **0.70** | **0.67** |
| | | $a_{FR}$ | 0.92 | 0.24 | 0.78 | 0.72 | 0.88 | 0.67 | 0.72 | **0.70** | 0.65 |
| | Bas. | $a_E$ | 0.89 | 0.44 | 0.72 | 0.76 | 0.85 | 0.67 | 0.65 | 0.76 | 0.59 |
| | | $a_{MSP}$ | 0.87 | 0.44 | 0.73 | 0.78 | 0.83 | 0.67 | 0.66 | 0.72 | 0.56 |
| | | $a_L$ | 0.78 | 0.79 | 0.61 | 0.74 | 0.92 | 0.66 | 0.65 | 0.95 | 0.51 |
| $s_1$ | Ours | $a_{D^*}$ | 0.89 | 0.34 | 0.60 | **0.87** | **0.51** | **0.71** | **0.85** | **0.52** | **0.59** |
| | | $a_{FR^*}$ | 0.86 | 0.37 | 0.61 | 0.65 | 0.93 | 0.64 | 0.77 | 0.77 | 0.58 |
| | | $a_{D_\alpha^{mean}}$ | 0.84 | 0.41 | 0.61 | 0.68 | 0.84 | 0.69 | 0.64 | 0.86 | 0.55 |
| | | $a_{D_\alpha^{idf}}$ | 0.82 | 0.48 | 0.61 | 0.66 | 0.87 | 0.69 | 0.63 | 0.87 | 0.55 |
| | | $a_{FR^{mean}}$ | 0.72 | 0.72 | 0.62 | 0.53 | 0.94 | 0.66 | 0.58 | 0.94 | 0.57 |
| | | $a_{FR^{idf}}$ | 0.62 | 0.85 | 0.67 | 0.55 | 0.96 | 0.58 | 0.56 | 0.93 | 0.57 |
| | Bas. | $a_M$ | **0.91** | **0.28** | **0.60** | 0.71 | 0.76 | 0.57 | 0.78 | 0.55 | 0.57 |
| | | $a_C$ | 0.70 | 0.80 | 0.63 | 0.65 | 0.76 | 0.64 | 0.71 | 0.61 | 0.48 |

Table 17: Summary of the results with additional finetuning of the models on the reference set. The results are similar to the results without finetuning as expected since the models had been trained initially on similar distributions.

