# OpenReview forum: "RainProof: An Umbrella to Shield Text Generator from Out-Of-Distribution Data"
_ICLR.cc/2023/Conference — Submitted to ICLR 2023_

### Official Review · Reviewer_JtYo · 2022-10-24

**Confidence:** 4
**Correctness:** 3
**Technical Novelty And Significance:** 3
**Empirical Novelty And Significance:** 3
**Recommendation:** 5

**Clarity, Quality, Novelty And Reproducibility:**

The clarity of the paper is good. The quality and novelty of the paper are fair. The experimental parts seem reproducible.

**Strength And Weaknesses:**

Strength:

1. The author adopts the OOD into the scenarios of text generation, which is a new task that lacks exploration.

2. The paper is clearly stated and easy to follow.

Weaknesses:

1. Some details of the proposed method lack theoretical justification. In equation (5), the author takes the average of the soft-distribution probabilities as a representation of the overall sentence distribution. This averaging step can lose a lot of sequential information of the input x. I am not convinced that the mean of the next-token prediction probabilities can represent the distribution information of the whole sentence in language modeling. In contrast, the token-level information measurements in equation (3) seem more reasonable to me. Why cannot the author define a token-level information measurement for the reference scenario, which might be consistent with the no-reference case?

2. The sequential likelihood method is not compared for the with-reference scenario. With the reference, one can fine-tune the generator to obtain a better sequential likelihood estimation for text. The estimated likelihood can be also regarded as a score of occurrence for input sequences. Is there any specific reason why the author did not compare with this simple baseline?

**Summary Of The Paper:**

This paper proposes an out-of-distribution (OOD) detection method for text generation based on the concept of information measure. More specifically, the author considers two measurements, the Renyi divergence, and the Fisher-Rao distance under both the no-reference and the reference scenarios. The empirical results show improvements with the proposed method in both dialogue generation and machine translation.

**Summary Of The Review:**

The paper designed several out-of-distribution detection methods for text generation based on information measurements, which is a good exploration of the OOD problem in text generation. However, some method details require more theoretical justification.

---

> ### Author Response · Authors · 2022-11-12
> **Answers to JtYo's comments**
>
> We thank JtYo for their reading of the paper and are glad to read that they acknowledge the novelty of the tasks investigated and the  clarity of the paper.
>
> In the following, we address in detail the issues raised by the reviewer:
>
> 1. **About the choice of aggregating the probability distributions** over the sequence before taking the information measurement, the main reason for doing so is due to the computational nature. We rely on the information projection onto the reference set which is an expensive operation. If we were to do it at each step as proposed by JtYo, we  would increase the size of the reference set by roughly two orders of magnitude. Since we would have to store reference distributions at each step.  It would increase the memory footprint too much (it would reach around 200Go). Moreover, at inference time, it would mean to compute the projection onto this enormous set at each step of the generation. We believe that it would be completely impractical. Putting aside those considerations, and as surprising as it may come, aggregating the probability distributions over the sequence by taking the mean has proved to be quite effective in sentence similarity tasks [Colombo, & al (2022). InfoLM: A New Metric to Evaluate Summarization & Data2Text Generation].
> 2. **We did not ourselves finetune the models. However, we did choose pretrained models distributed with validation and test data,  and we used the test data as IN samples we wanted our detectors to keep.** Therefore, the models have been fine tuned.  Fine tuning further on the validation set is not something we considered but for comprehensivity sake we did reiterate our experiments with additional finetuning and present those results in Table 17. They are very similar to our current results and do not show any significant changes. We also want to point out that we wanted to avoid additional finetuning in order to propose practical, easy to set up, plug&play methods.
>
> We did our best to answer each concern of reviewer JtYo and believe that we managed to improve enough our work based on their suggestions to ask them to reconsider our paper. We hope reviewer JtYo is satisfied by our answers and revision of the paper, we would like them to reconsider their grade.

---

### Official Review · Reviewer_BFEH · 2022-10-24

**Confidence:** 4
**Correctness:** 3
**Technical Novelty And Significance:** 3
**Empirical Novelty And Significance:** 3
**Recommendation:** 6

**Clarity, Quality, Novelty And Reproducibility:**

The work targets distribution detection for generation tasks, an area that has not been explored much as most previous work has focused on classification tasks. The ideas are novel and have been explained properly in the paper. More implementation details would help in the reproducible aspect.

**Strength And Weaknesses:**

1. A major strength is that detecting the distribution does not hamper the performance of the model and is task independent.

2. It is fully unsupervised, thus there is no need for annotated data.

3. For similarity calculation, it would be good to provide results on how well the model does when the common similarity metrics like cosine and dot product is used.

4. It would be good to see the performance of other reference distribution for u.

5. Why is F1 score not considered along with AUROC.

**Summary Of The Paper:**

The authors have provided an out of distribution detector, RAINPROOF and an out of distribution performance benchmark, LOFTER. RAINPROOF is an unsupervised model that is able to detect whether a reference sample is in or out of distribution when compared to the training distribution, without hampering the overall performance of the system.

They find the similarity score between two samples and apply a threshold to classify it as in or out of distribution. During inference, a soft distribution is extracted and used to compute an anomaly score, which is thresholded to perform the in or out of distribution classification.

The metrics used were AUROC and FPR and the authors report better performance on domain and dialog shifts. In language shifts, the performance is mixed, as the baselines perform better for , which is the data-driven notion normality.

**Summary Of The Review:**

There are some typos in the paper. Particularly on line 10 of the abstract (ODD -> OOD) and section 3.3 has some grammatical errors and typos:
1. In the with reference scenario -> In the reference scenario
2. weakness of eq 3 is to imposes -> weakness of eq 3 is that it imposes
3. under s1, our OOD detecor -> our OOD detector
4. the last step consists in -> the last step consists of

Some experimental results on the choice of similarity metric and reference distribution would help to understand the current selections.

---

> ### Author Response · Authors · 2022-11-12
> **Answers to BFEH's comments**
>
> We would like to thank BFEH for its detailed reading of the manuscript. We are glad that they acknowledge that our proposal brings significant contributions and enables efficient OOD detection as well as improvement of task specific performance.
>
> In the following, we provide detailed answers to each issue and remarks raised by BFEH:
>
> 1. **About cosine similarities,** we chose to work in a black-box setting assuming that we did not have access to the hidden representations. We only included the Mahalanobis distance since it is the most common features-based OOD detectors and a common baseline. It is worth pointing out that when working with features based detectors, we need to decide which layer of the encoder to consider. We wish to avoid as much as possible any manual features selection. Following the review, we revised our experiments and added the cosine similarity as baseline in the reference scenario leading to similar results as the Mahalanobis distance.
>
> 2. **About other reference distributions,** we worked on other options of reference distributions. We kept specifically the uniform distribution as reference since it yields the negentropy [Brillouin, Leon: (1953) "Negentropy Principle of Information"].
>
>     We can first point out that the **likelihood can be seen as the divergence between the probability distribution and the dirac distribution of the actually chosen token.** For sake  of comprehensivenes, we added  results obtained by using the IDF distribution  [Yuan, W., Neubig, G., & Liu, P. (2021). Bartscore: Evaluating generated text as text generation]  over the reference set and simply the average probability distribution obtained on the reference set. However, these methods become “with reference” since we somehow need to compute the reference distributions from a reference set, whereas the negentropy gives a notion of normality that is not data driven.
>
>
>     We provide these additional results in Tables 16 and 17. They do not change the initial conclusions and  support our results showing that the uncertainty of the model, measured in terms of negentropy, provides better insights about its behavior and performance than comparing new samples to a reference set of known IN samples.
>
> 3. **We did consider the F1 score**, we just did not add it for sake of brevity. Following the reviewer’s remark, we added it to the paper.
> 4. **About reproducibility and implementation details, **we will release upon acceptance a python library built on top of Huggingface transformers [Thomas Wolf, et al.. 2020. Transformers: State-of-the-Art Natural Language Processing.] dedicated to OOD detection for text generation and selective generation** that will implement all the results and detectors presented in the paper.** More specifically, it implements tools and syntax to add a detector on top of any generative process in a plug and play manner. We are still in the process of tailoring and documenting precisely the code to ensure its usability.
>
> We hope that we addressed all concerns made by reviewer BFEH and revised our paper accordantly. We hope that the reviewer  will be keen to consider raising their score accordingly.

---

### Official Review · Reviewer_PSuw · 2022-10-28

**Confidence:** 4
**Correctness:** 3
**Technical Novelty And Significance:** 2
**Empirical Novelty And Significance:** 2
**Recommendation:** 3

**Clarity, Quality, Novelty And Reproducibility:**

The motivation is clear.
The quality could be further improved.
Novelty is limited.

Suggestion:
- In section 4.2, it's unnecessary to add an arrow after every AUROC and FPR.
- The notation part is distributed in two places, 2.1/2.2. Similarly, the related work is also appeared in two places 2.3/3.1. It will be easier to read if related work are together and important notations are gathered.

**Strength And Weaknesses:**


Lack of baselines
I think the experiment lacks an important baseline, the perplexity (PPL), since this work is based on large pretrained language model. The language model is originally designed to maximize the $p(x) = p(x_1)p(x_2)...p(x_n)$, which is a great anomaly detector itself.

Experiments
- Have you ever test your model on some other classical OOD tasks?

Problem
- Could the author give more explanation on no-referene scenario? Take mutual information, a special case of Renyi divergence as an example, I(P, U) = H(P) + H(U) because U and P are independent variables. H(U) is a constant variable for any distribution.  As the results, the entropy is used as the novelty score.
- Did you fine-tune the language-model with in-domain data?


Performance
- From the reviewers' experience on OOD in text, the performance could be much better using a simple plug-in structure. (The reviewer understood that the authors want to avoid the extra plug-in structure, just curious if better results can be achieved)


**Summary Of The Paper:**

The paper explored OOD detection in text. Different from previous OOD work, this paper more focuses on realistic data shifts and domain change. To measure the out-of-domain score, the authors introduce renting divergence and fisher-ran distance.

**Summary Of The Review:**

I have some doubts on the methods. The experiment part should compare with more baselines on some common OOD datasets. Currently, nearly all the datasets are synthesized by authors and there is no demo in supplementary.

---

> ### Author Response · Authors · 2022-11-12
> **Answers to PSuw comments**
>
> First of all, we thank PSuw for their careful reading of the manuscrit.
>
> In the following, we provide detailed answers for PSuw’s issues and remarks:
>
> 1. **About the lack of baselines:** PSuw points that we should have included the perplexity. We did use and report in the paper the **log-likelihood** of the generated sequences which is effectively the perplexity. Indeed, the perplexity is exactly the likelihood $L(x) = \prod_{t=1)^T p(x_{t+1} | x_{\leq t})}$ of the sequence according to the language model. For sake of clarity, we added a footnote explaining the link between perplexity and log-likelyhood and it is further detailed in Appendix A.6.
> 2. **About the other “classical OOD tasks”:** Our work addresses the problem of OOD detection for text generation which is different from textual classification (the standard setting for OOD detection, see [Tan, Ming, et al. "Out-of-domain detection for low-resource text classification tasks.", Jeong, Taewon, and Heeyoung Kim. "OOD-MAML: Meta-learning for few-shot out-of-distribution detection and classification." ]).
>
> As stated in the paper, we chose to tackle the problem of text generation because it was largely ignored and most techniques focused on classification tasks. Therefore, it is not clear to us what would be a classical OOD task in the present scenario.
>
>
> **Issues raised**
>
>
> 1. **About the no-reference scenario:**
>
> We would like to clarify that **the mutual information between random variables P and U is radically different from the Renyi divergence between P and U**. The mutual information cannot be shown to be as a special case of Renyi divergence. The mutual information is to be applied between two random variables X, Y and corresponds to the Kullback-Leibler divergence between the actual joint distribution of X and Y and the product distribution of the two variables.
>
> $I(X ; Y) = D_{KL}(P_{X, Y} || P_X \times P_Y).$
>
> In our case, we only have the two marginal distributions but we don’t have access to the joint distribution.  Moreover, it appears that  **statement is incomplete:** if we have X and U two random variables,
>
> $I(X ; U) = H(X) + H(U) - H(X, U)$ where the term $H(X, U)$ remains unknown. If X and U are independent random variables, as suggested by the reviewer, then      $I(X ; U) = 0$.
>
> Please also note that in our paper we do not use or mention mutual information. What we do instead is to compute the negentropy [Brillouin, Leon: (1953) "Negentropy Principle of Information"]. As stated in the paper, the alpha Renyi-negentropy of a r.v X following P is obtained by computing the alpha-Renyi-divergence between P and the uniform distribution.
>
> In the special case of the limit when $alpha \rightarrow  1$, a connection to the Shannon Entropy can be obtained but it also includes a regularization term depending on the size of the support of the distribution (i.e., the size of the non zero probability token). We refer the reviewer to the paper for further details (see Section 3.2).
>
> 2. **About fine-tuning of the models:** we chose pretrained models distributed according to the validation and test data. We used the test data as IN samples we wanted our detectors to preserve as normal. Therefore, the models have been fine tuned, although not by us.  Following reviewer JtYo and reviewer PSuw, we did perform additional finetuning on the validation set and present those results in Table 17. They are very similar to our current results and do not show any significant changes. We also want to point out that we preferred  to avoid additional finetuning in order to propose practical, easy to set up, plug&play methods.
>
> **Questions:**
>
> **About an additional plugin structure.** We understand that PSuw suggests plugging a trainable model to the output to distinguish OOD samples. As acknowledged by the reviewer, we voluntarily decided not to pursue this option because it induces a training overhead that we believe is not practical. In addition, recent work on the learnability of OOD detection proves several impossibility theorems in the PAC framework (under $s_0$) [Fang, Zhen et al. “Is Out-of-Distribution Detection Learnable?”]. However, if the reviewer could provide a reference, we would be happy to mention the study suggested by the reviewer on plug-in structures for future research directions.
>
> In summary, we hope we have properly addressed the PSuw reviewer's concerns because (1) the proposed baseline is already included in our paper although under a different name; (2) the models have already been fine-tuned on the IN distribution data and further refinements did not change the results;  (3) the question on mutual information  arises from a misunderstanding; and (4) we have made changes to clarify the technical parts.
>
> We hope our answer will alleviate PSuw concerns and we kindly ask reviewer PSuw to reconsider their opinion on the paper and review their grade.

---

### Author Response · Authors · 2022-11-12
**General answer**

# Global comment:

We want to warmly thank all the reviewers for their work and careful assessment of our paper. We are glad that they acknowledge the clarity of the motivation and **the novelty  of addressing OOD detection in the generative scenario** (Reviewer JtYo and Reviewer BFEH). We invested our best efforts to answer and to improve every issue raised by the reviewers. We have updated the paper accordingly and highlighted the changes made.

In particular, we updated the manuscript to answer the reviewers’ criticisms regarding:

1. The typos and other remarks on the form ;
2. The metrics: Reviewer BFEH asked for F1 score, we added it along with precision and recall ;
3. An additional baseline requested  by Reviewer BFEH and Reviewer PSuw (namely fine-tuning the models and cosine similarities) ;
4. We have also updated parts about the reference scenario, especially why not taking a projection score at each step instead of aggregating the probability distributions (e.g., as suggested by reviewer JtYo).

**Regarding the additional baseline.**

In all our experiments, we used pre-trained models. We worked with IN distribution datasets that are part of the pre-training data. Therefore, the models were already fine-tuned on the IN distribution data. However, following the suggestion of the reviewers, we performed additional experiments and further fine-tuned them on the validation set. The results are displayed in Table 17. As the reviewers can see, there is no change in our conclusions.

Below, we provided a detailed  answer to all reviewer's comment and included their valuable suggestions into our work.

---

### Decision · Program_Chairs · 2023-01-20

**Decision:**

Reject

**Justification For Why Not Higher Score:**

While OOD for text generation is relatively unexplored, I think the authors could have made some efforts to make some indirect comparisons. Lack of these makes it very difficult for the reviewers and the chair to correctly assess the paper.

**Justification For Why Not Lower Score:**

N/A

**Metareview: Summary, Strengths And Weaknesses:**

Summary: The paper is tackling OOD detection in text space. There is no doubt that this is a very important problem in the field. However, the datasets used in this paper are new in the paper and thus cannot be easily compared with previous papers. It is true that most previous work is text classification based, but still, given that text classification can be considered as a subset of text generation problems, including the result would have been better. Overall, while the authors have addressed many of the issues raised, at the end, the paper as a whole does not seem to be robust enough to be presented at ICLR at the moment.

Strengths:
- All reviewers agree that the motivation of the paper is strong.
- Some reviewers note the unique benefit of the proposed model.

Weaknesses:
- Some reviewers are concerned that the paper lacks simple baselines.
- Some reviewers question the effectiveness of the proposed method, especially given that the datasets used are new in the paper.